# Minimax Optimal Nonparametric Estimation of Heterogeneous Treatment Effects

**Zijun Gao**
Department of Statistics
Stanford University
Email: zijungao@stanford.edu

**Yanjun Han**
Department of Electrical Engineering
Stanford Univeristy
Email: yjhan@stanford.edu

## Abstract

A central goal of causal inference is to detect and estimate the treatment effects of a given treatment or intervention on an outcome variable of interest, where a member known as the heterogeneous treatment effect (HTE) is of growing popularity in recent practical applications such as the personalized medicine. In this paper, we model the HTE as a smooth nonparametric difference between two less smooth baseline functions, and determine the tight statistical limits of the nonparametric HTE estimation as a function of the covariate geometry. In particular, a two-stage nearest-neighbor-based estimator throwing away observations with poor matching quality is near minimax optimal. We also establish the tight dependence on the density ratio without the usual assumption that the covariate densities are bounded away from zero, where a key step is to employ a novel maximal inequality which could be of independent interest.

## 1 Introduction and Main Results

Causal inference aims to draw a causal relationship between some treatment and target responses. Nowadays, personalized medicine and huge available data make heterogeneous treatment effect (HTE) estimation meaningful and possible. While there are various practical approaches of estimating the HTE [AI16, PQJ+17, WA18, KSBY19], some important theoretical questions remain unanswered.

In this paper, we consider the Neyman-Rubin potential outcome model [Rub74] for the treatment effect. Assume for simplicity that there are $n$ individuals in the treatment group and the control group, respectively, where the generalizations to different group sizes are straightforward. For each individual $i \in [n]$ in the control group, we observe a vector of covariates $X_i^0 \in \mathbb{R}^d$ and her potential outcome $Y_i^0 \in \mathbb{R}$ for not receiving the treatment. Similarly, for individual $i$ in the treatment group, the covariates $X_i^1$ and the potential outcome $Y_i^1$ under the treatment are observed. We assume the following model for the potential outcomes: for any $i \in [n]$,

$$Y_i^0 = \mu_0(X_i^0) + \varepsilon_i^0, \qquad Y_i^1 = \mu_1(X_i^1) + \varepsilon_i^1, \tag{1}$$

where $\mu_0, \mu_1 : \mathbb{R}^d \to \mathbb{R}$ are the baseline functions for the control and treatment groups, respectively, and $\varepsilon_i^0, \varepsilon_i^1$ are modeling errors. The heterogeneous treatment effect $\tau$ is defined to be the difference of the baseline functions:

$$\tau(x) \triangleq \mu_1(x) - \mu_0(x). \tag{2}$$

In other words, the treatment effect $\tau(x)$ is the expected change in the outcomes after an individual with covariate $x$ receives the treatment, which is usually heterogeneous as $\tau(x)$ typically varies in $x$. The target is to find an estimator $\hat{\tau}$ which comes close to the true HTE $\tau$ under the $L_2$ norm of functions, based on the control and treatment observations $\{(X_i^0, Y_i^0)\}_{i \in [n]}, \{(X_i^1, Y_i^1)\}_{i \in [n]}$.

In HTE estimation, modeling of the baseline functions $\mu_0, \mu_1$ or the HTE $\tau$ plays an important role. In practice, the treatment effect $\tau$ is typically easier to estimate than the baseline functions $\mu_0, \mu_1$, as ideally the treatment effect depends solely on the single treatment. In this paper, we assume that both the baseline and treatment effect functions are nonparametric functions, with an additional constraint that $\tau$ is smoother than $\mu_0$:

**Assumption 1** (Baseline and HTE functions). *The baseline $\mu_0$ and the treatment effect $\tau$ belong to d-dimensional Hölder balls with smoothness parameters $\beta_\mu \leq \beta_\tau$, respectively.*

Recall the following definition of Hölder balls.

**Definition 1** (Hölder ball). *The Hölder ball $\mathcal{H}_d(\beta)$ with dimension $d$ and smoothness parameter $\beta \geq 0$ is the collection of functions $f : \mathbb{R}^d \to \mathbb{R}$ supported on $[0,1]^d$ with*

$$\left| \frac{\partial^s f}{\partial x_1^{\beta_1} \cdots \partial x_d^{\beta_d}}(x) - \frac{\partial^s f}{\partial x_1^{\beta_1} \cdots \partial x_d^{\beta_d}}(y) \right| \leq L\|x-y\|_2^\alpha$$

*for all $x, y \in \mathbb{R}^d$ and all multi-indices $(\beta_1, \cdots, \beta_d) \in \mathbb{N}^d$ with $\sum_{i=1}^d \beta_i = s$, where $\beta = s + \alpha$ with $s \in \mathbb{N}, \alpha \in (0,1]$. Throughout we assume that the radius $L$ is a fixed positive constant and omit the dependence on $L$.*

Assumption 1 imposes no structural assumptions on $\mu_0$ and $\tau$ except for the smoothness, and assumes that the HTE $\tau$ is smoother and thus easier to estimate than the baseline $\mu_0$. This typically holds as intuitively fewer factors contribute to the HTE than baselines, and semiparametric models usually assume simple forms of HTEs such as constant or linear [Ken16, CCD+18], where Assumption 1 could be treated as a nonparametric counterpart. Standard results in nonparametric estimation reveal that if one naïvely estimates the baselines $\mu_0$ and $\mu_1$ separately based on (1), then each function can be estimated within accuracy $\Theta(n^{-\beta_\mu/(2\beta_\mu+d)})$, and so is the difference $\tau$ in (2). However, if the covariates in the control and treatment groups match perfectly, i.e. $X_i^0 = X_i^1$ for all $i \in [n]$, then differencing (1) gives an improved estimation accuracy $\Theta(n^{-\beta_\tau/(2\beta_\tau+d)})$ for the HTE $\tau$. In general, the estimation performance of HTE is an interpolation between the above extremes and depends heavily on the quality of the covariate matching. To model such qualities, we have the following assumption on the covariates $X_i^0, X_i^1$.

**Assumption 2** (Covariates). *The covariates are generated under fixed or random designs below:*

- Fixed design: *the covariates are generated from the following fixed grid*

$$\left\{X_i^0\right\}_{i \in [n]} = \left\{0, \frac{1}{m}, \cdots, \frac{m-1}{m}\right\}^d, \quad \left\{X_i^1\right\}_{i \in [n]} = \left\{0, \frac{1}{m}, \cdots, \frac{m-1}{m}\right\}^d + \Delta,$$

  *for some vector $\Delta \in \mathbb{R}^d$ with $\|\Delta\|_\infty \leq 1/(2m)$, and $m \triangleq n^{1/d}$ is assumed to be an integer.*

- Random design: *the covariates are i.i.d. sampled from unknown densities $g_0, g_1$ on $[0,1]^d$:*

$$X_1^0, X_2^0, \cdots, X_n^0 \overset{\text{i.i.d.}}{\sim} g_0, \quad X_1^1, X_2^1, \cdots, X_n^1 \overset{\text{i.i.d.}}{\sim} g_1,$$

  *where there is a bounded likelihood ratio $\kappa^{-1} \leq g_0(x)/g_1(x) \leq \kappa$ on the densities.*

Under the fixed design, the covariates in both groups are evenly spaced grids in $[0,1]^d$, with a shift $\Delta$ quantifying the matching distance between the control and treatment groups. The fixed design is not very practical, but the analysis will provide important insights for the HTE estimation. The random design model is more meaningful and realistic without any matching parameter, and the density ratio $g_0/(g_0 + g_1)$ corresponds to the propensity score, a key quantity in causal inference. We assume that $\kappa^{-1} \leq g_0/g_1 \leq \kappa$ everywhere, as it is usually necessary to have a propensity score bounded away from 0 and 1. Besides the bounded likelihood ratio, we remark that it is *not* assumed that the densities $g_0, g_1$ are bounded away from zero.

Finally it remains to model the noises $\varepsilon_i^0, \varepsilon_i^1$, and we assume the following mild conditions.

**Assumption 3** (Noise). *The noises $\varepsilon_i^0, \varepsilon_i^1$ are mutually independent, zero-mean and of variance $\sigma^2$.*

Based on the above assumptions, the target of this paper is to characterize the minimax risks of the nonparametric HTE estimation under both fixed and random designs. Specifically, we are interested in the following minimax $L_1$ risk

$$R_{n,d,\beta_\mu,\beta_\tau,\sigma}^{\text{fixed}}(\Delta) \triangleq \inf_{\hat{\tau}} \sup_{\mu_0 \in \mathcal{H}_d(\beta_\mu), \tau \in \mathcal{H}_d(\beta_\tau)} \mathbb{E}_{\mu_0,\tau}[\|\hat{\tau} - \tau\|_1]$$

for fixed design with matching parameter $\Delta \in \mathbb{R}^d$, where the infimum is taken over all possible estimators $\hat{\tau}$ based on the observations $\{(X_i^0, Y_i^0)\}_{i \in [n]}, \{(X_i^1, Y_i^1)\}_{i \in [n]}$. As for the minimax risk for random designs, we include the dependence on the density ratio $\kappa$ and use the $L_1$ norm based on the density $g_0$ (as we do not assume that the densities are bounded away from zero):

$$R_{n,d,\beta_\mu,\beta_\tau,\sigma}^{\text{random}}(\kappa) \triangleq \inf_{\hat{\tau}} \sup_{\mu_0 \in \mathcal{H}_d(\beta_\mu), \tau \in \mathcal{H}_d(\beta_\tau), g_0, g_1} \mathbb{E}_{\mu_0,\tau}[\|\hat{\tau} - \tau\|_{L_1(g_0)}].$$

The main aim of this paper is to characterize the tight minimax rates for the above quantities, and in particular, how they are determined by the covariate geometry and the full set of parameters $(n, \kappa, \sigma)$ of interest. In addition, we aim to extract useful practical insights (instead of fully practical algorithms) based on the minimax optimal estimation procedures established in theory. Moreover, we mostly focus on the special nature of the HTE estimation problem instead of general nonparametric estimation, and therefore we elaborate less on broad issues of nonparametric statistics such as adaptation/hyperparameter estimation and refer interested readers to known literature (see Section 5).

Our first result is the characterization of the minimax rate for HTE estimation under fixed designs.

**Theorem 1** (Fixed Design). *Under Assumptions 1–3,*

$$R_{n,d,\beta_\mu,\beta_\tau,\sigma}^{\text{fixed}}(\Delta) \asymp n^{-\frac{\beta_\mu}{d}} (n^{\frac{1}{d}} \|\Delta\|_\infty)^{\beta_\mu \wedge 1} + \left(\frac{\sigma^2}{n}\right)^{\frac{\beta_\tau}{2\beta_\tau + d}}.$$

Theorem 1 shows that, as the covariate matching quality improves (i.e. $\|\Delta\|_\infty$ shrinks), the estimation error of the HTE decreases from $n^{-\beta_\mu/d}$, which is slightly better than the estimation error $n^{-\beta_\mu/(2\beta_\mu+d)}$ for the baselines, to the optimal estimation error $(\sigma^2/n)^{\beta_\tau/(2\beta_\tau+d)}$ when the learner has $n$ direct samples from $\tau$. We also remark that the covariate matching quality is determined by the $\ell_\infty$ norm of $\Delta$, and the matching bias does not depend on the noise level $\sigma$.

The minimax rate for HTE estimation under random designs exhibits more interesting behaviors.

**Theorem 2** (Random Design). *Under Assumptions 1–3, if $\beta_\tau \leq 1$ and $\kappa \leq n$, then*

$$R_{n,d,\beta_\mu,\beta_\tau,\sigma}^{\text{random}}(\kappa) = \widetilde{\Theta}\left(\left(\frac{\kappa}{n^2}\right)^{\frac{1}{d(\beta_\mu^{-1} + \beta_\tau^{-1})}} + \left(\frac{\kappa\sigma^2}{n^2}\right)^{\frac{1}{2+d(\beta_\mu^{-1} + \beta_\tau^{-1})}} + \left(\frac{\kappa\sigma^2}{n}\right)^{\frac{\beta_\tau}{2\beta_\tau + d}}\right).$$

Theorem 2 provides the first minimax analysis of the HTE estimation with tight dependence on all parameters $(n, \kappa, \sigma)$ without assuming densities bounded away from zero. Interestingly, Theorem 2 shows that there are three regimes of the minimax rate of the HTE estimation under random designs, and in particular, there is an intermediate regime where neither the matching bias nor the estimation of $\tau$ dominates. Moreover, when it comes to the dependence on the density ratio $\kappa$, the effective sample size is $n/\sqrt{\kappa}$ in the first two regimes, while it becomes $n/\kappa$ in the last regime.

## 1.1 Related work

The nonparametric regression of a single function has a long history and is a well-studied topic in nonparametric statistics, with various estimators including kernel smoothing [Ros56], local polynomial fitting [Sto77, FGG+97], splines [DBDBM+78, GS93], wavelets [DJKP95], nearest neighbors [Fix51, Sto77], and so on. We refer to the excellent books [Nem00, GKKW06, Tsy09, Joh11, BD15] for an overview. However, estimating the difference of two nonparametric functions remains largely underexplored.

In the history of causal inference, the average treatment effect (ATE) has long been the focus of research [IR15]. Recently, an increasing number of works seek to estimate the HTE, and there are two major threads of approaches: parametric (semi-parametric) estimation and nonparametric estimation.

In parametric estimation, a variety of parametric assumptions such as linearity are imposed on the baseline functions and HTE. The problem then reduces to classic parametric estimation and many methods are applicable, e.g. [TAGT14, IR$^+$13]. In this paradigm, classic parametric rate is obtained under proper regularity assumptions. To further deal with observational study and bias due to the curse of high-dimensionality, [CCD$^+$18] develops a double/debiased procedure allowing a less accurate estimation of the mean function without sacrificing the parametric rate.

In nonparametric estimation, there are a large number of practical approaches proposed, such as spline-based method [Che07], matching-based method [XBJ12] and tree-based method [AI16, PQJ$^+$17, WA18]. However, relatively fewer work study the statistical limit of nonparametric estimation of HTE. One work [NW17] argued that a crude estimation of baselines suffices for the optimal estimation of HTE, while their approach did not take into account the impact of the covariate geometry. Some works [AS18, KSBY19] studied the minimax risk with smoothness assumptions on the baseline functions, but they did not directly model the smoother HTE function and thus arrived at a relatively easier claim that the difficulty is determined by the less smooth baseline. In contrast, we assume a smoother HTE and provide additional insights on how to construct pseudo-observations and discarding poor-quality data based on the covariate geometry. Hence, to the best of our knowledge, this paper establishes the first minimax rates with tight dependence on both the covariate geometry, and the parameters $\kappa$ related to propensity scores as well as $\sigma$ related to noise levels.

We also comment that in nonparametric regression problems with random design, a uniform lower bound on the density is usually necessary in most past works [BCLZ02, GKKW06]. To overcome this difficulty, there is a recent line of research showing that to estimate nonparametric functionals, Hardy–Littlewood-type maximal inequalities are often useful to deal with densities close to zero for both kernel methods [HJWW17] and nearest neighbors [JGH18]. Therefore, this work is a continuation of the above thread with a novel maximal inequality on another different problem.

## 1.2 Notations

Let $\mathbb{R}, \mathbb{N}$ be the set of all real numbers and non-negative integers, respectively. For $p \in [1, \infty]$, we denote by $\| \cdot \|_p$ both the $\ell_p$ norm for vectors and the $L_p$ norm for functions. Let $[n] \triangleq \{1, 2, \cdots, n\}$, and $a \wedge b \triangleq \min\{a, b\}$. We call $K : \mathbb{R}^d \to \mathbb{R}$ a kernel of order $k$ if $\int K(x)dx = 1$ and $\int (v^\top x)^\ell K(x)dx = 0$ for any $v \in \mathbb{R}^d$ and $1 \leq \ell \leq k$. For non-negative sequences $\{a_n\}$ and $\{b_n\}$, we write $a_n \asymp b_n$, or $a_n = \Theta(b_n)$, to denote that $cb_n \leq a_n \leq Cb_n$ for some absolute constants $c, C > 0$ independent of $n$. The notation $a_n = \widetilde{\Theta}(b_n)$ means that the above inequality holds within multiplicative polylogarithmic factors in $n$.

## 1.3 Organization

In Section 2, we construct the minimax optimal HTE estimator via a combination of kernel methods and covariate matching under the simple fixed design setting. For the random design, a two-stage nearest-neighbor based algorithm is proposed in Section 3 to trade off two types of biases and the variance. The efficacy of the proposed estimator is shown via numerical experiments in Section 4. Some limitations and future works are discussed in Section 5, and all proofs are relegated to the Appendices.

## 2 Fixed Design

This section is devoted to the minimax rate of HTE estimation under fixed design. To construct the estimator, we first form pseudo-observations of the outcomes in the treatment group on the covariates in the control group based on covariate matching, and then apply the classic kernel estimator based on perfectly matched pseudo-observations. We also sketch the ideas of the minimax lower bound and show that both the matching bias and the estimation error are inevitable.

### 2.1 Estimator Construction

Recall that in the perfect matching case, i.e. $\Delta = 0$, the HTE function $\tau$ can be estimated by classic nonparametric estimators after taking the outcome difference $Y_i^1 - Y_i^0$ in (1). This basic idea will be

---
**Algorithm 1** Estimator Construction under Fixed Design
---
**Input:** control observations $\{(X_i^0, Y_i^0)\}_{i \in [n]}$, treatment observations $\{(X_i^1, Y_i^1)\}_{i \in [n]}$, dimension-ality $d$, kernel $K$ of order $\lfloor \beta_\tau \rfloor$, bandwidth $h > 0$, query $x_0 \in [0,1]^d$.
**Output:** HTE estimator $\hat{\tau}(x_0)$ at point $x_0$.
**for** $x \in \{X_i^0\}_{i \in [n]}$ **do**
    Choose $t = \lfloor \beta_\mu \rfloor + 1$, and find neighboring treatment covariates $\{x_i\}_{i \in [t]^d}$;
    Compute the weights $\{w_i\}_{i \in [t]^d}$ according to (3);
    Compute the pseudo-observation $\hat{Y}^1(x)$ according to (4).
**end for**
Output the final estimator $\hat{\tau}(x_0)$ according to (5).
---

used for general fixed designs: first, for each $i \in [n]$, we form pseudo-observations $\hat{Y}_i^1$ with target mean $\mu_1(X_i^0)$, where the observed covariate $X_i^1$ in the treatment group moves to the nearest covariate $X_i^0$ in the control group. Next, we apply the kernel estimator in the perfect matching case to the pseudo-difference $\hat{Y}_i^1 - Y_i^0$. The covariate matching step also makes use of a suitable kernel method, where the matching performance depends on the smoothness of the baseline function $\mu_0$.

Specifically, the pseudo-observations $\hat{Y}_i^1$ are constructed as follows. Let $t \triangleq \lfloor \beta_\mu \rfloor + 1$, and fix a covariate $x$ in the set of control covariates. For each $j \in [d]$, let $x_{1,j}, \cdots, x_{t,j}$ be the $t$ closest grid points in $\{\Delta_j, 1/m + \Delta_j, \cdots 1 - 1/m + \Delta_j\}$ to $x_j$, the $j$-th coordinate of $x$. Moreover, for each coordinate $j \in [d]$, we also compute the following weights $w_{1,j}, \cdots, w_{t,j}$ such that

$$\sum_{i=1}^{t} w_{i,j}(x_{i,j} - x_j)^\ell = \mathbb{1}(\ell = 0), \quad \forall 0 \le \ell \le \lfloor \beta_\mu \rfloor. \tag{3}$$

To form the pseudo-observation of the treatment outcome $\hat{Y}^1(x)$ on the covariate $x$, we use the treatment observations $Y^1(x_i)$ for all multi-indices $i = (i_1, \cdots, i_d) \in [t]^d$ and $x_i \triangleq (x_{i,1}, x_{i,2}, \cdots, x_{i,d})$, with $w_i = w_{i,j}$ being the weight of the covariate $x_i$. Note that each $x_i$ belongs to the covariate grid of the treatment group and therefore $Y^1(x_i)$ is observable. Finally, given the above covariates $x_i$ and weights $w_i$, the pseudo-observation of the treatment outcome at the control covariate $x$ is

$$\hat{Y}^1(x) = \sum_{i \in [t]^d} w_i Y^1(x_i) \tag{4}$$

In approximation theory, this is an interpolation estimate of $\mu_1(x)$ based on the function values $\{\mu_1(x_i)\}_{i \in [t]^d}$ in the neighborhood of $x$.

Given the pseudo-observations, the final HTE estimator $\hat{\tau}(x_0)$ for any $x_0 \in [0,1]^d$ is defined as

$$\hat{\tau}(x_0) \triangleq \frac{\sum_{i=1}^{n} K((X_i^0 - x_0)/h)(\hat{Y}^1(X_i^0) - Y_i^0)}{\sum_{i=1}^{n} K((X_i^0 - x_0)/h)}, \tag{5}$$

i.e. the Nadaraya-Watson estimator [Nad64, Wat64] applied to the pseudo-differences, where $K$ is any kernel of order $\lfloor \beta_\tau \rfloor$, and $h \in (0,1)$ is a suitable bandwidth. The complete description of the estimator construction is displayed in Algorithm 1.

The performance of the above estimator is summarized in the following theorem.

**Theorem 3.** *There exists a constant $C > 0$ independent of $(n, h)$ such that*

$$\sup_{\mu_0 \in \mathcal{H}_d(\beta_\mu), \tau \in \mathcal{H}_d(\beta_\tau)} \mathbb{E}_{\mu_0, \tau}[\|\hat{\tau} - \tau\|_1] \le C \left( n^{-\frac{\beta_\mu}{d}} (n^{\frac{1}{d}} \|\Delta\|_\infty)^{\beta_\mu \wedge 1} + h^{\beta_\tau} + \frac{\sigma}{\sqrt{nh^d}} \right).$$

*In particular, the estimator $\hat{\tau}$ in (5) achieves the upper bound in Theorem 1 for $h = \Theta(n^{-1/(2\beta_\tau + d)})$.*

We sketch the proof idea of Theorem 3 here. Using the definition of pseudo-observations $\hat{Y}^1$, for each $i \in [n]$ we have $\hat{Y}^1(X_i^0) - Y_i^0 = \tau(X_i^0) + b_\mu(X_i^0) + \tilde{\varepsilon}_i$, where $\tau(X_i^0)$ is the target treatment effect at the point $X_i^0$, $b_\mu(x) = \sum_{i \in [t]^d} w_i \mu_1(x_i) - \mu_1(x)$ is the matching bias incurred by the linear interpolation, and $\tilde{\varepsilon}_i$ is a linear combination of the error terms. Then the first term of Theorem 3 is an upper bound of the matching bias, while the other terms come from the traditional bias-variance tradeoff in nonparametric estimation. We leave the full proof to the appendix.

## 2.2 Minimax Lower Bound

In this section, we show that the above HTE estimator is minimax optimal via the following minimax lower bound.

**Theorem 4.** *There exists a constant $c > 0$ independent of $n$ such that for any HTE estimator $\hat{\tau}$, it holds that under Gaussian noise,*

$$\sup_{\mu_0 \in \mathcal{H}_d(\beta_\mu), \tau \in \mathcal{H}_d(\beta_\tau)} \mathbb{E}_{\mu_0, \tau}[\|\hat{\tau} - \tau\|_1] \geq c \left( n^{-\frac{\beta_\mu}{d}} (n^{\frac{1}{d}} \|\Delta\|_\infty)^{\beta_\mu \wedge 1} + \left( \frac{\sigma^2}{n} \right)^{\frac{\beta_\tau}{2\beta_\tau + d}} \right).$$

As for the proof of Theorem 4, note that the estimation error $(\sigma^2/n)^{\beta_\tau/(2\beta_\tau + d)}$ is optimal in the nonparametric estimation of $\tau$ even if there is a perfect covariate matching. Hence, it remains to prove the first term, i.e. the matching bias. The proof is based on Le Cam's two-point method [Yu97] and a functional optimization problem. Consider two scenarios $(\mu_0, \tau)$ and $(\mu_0', \tau')$ with $\sigma = 0$ and $\mu_0'(x) \equiv 0, \tau'(x) \equiv 0$, where $(\mu_0(x), \tau(x))$ is the solution to the following optimization problem:

$$\begin{aligned}
\text{maximize} \quad & \|\tau\|_1 \\
\text{subject to} \quad & \mu_0(X_i^0) = 0, \quad \mu_0(X_i^1) + \tau(X_i^1) = 0, \quad i = 1, \cdots, n, \\
& \mu_0 \in \mathcal{H}_d(\beta_\mu), \tau \in \mathcal{H}_d(\beta_\tau).
\end{aligned} \tag{6}$$

Note that when $(\mu_0, \tau)$ is a feasible solution to (6), under both scenarios $(\mu_0, \tau)$ and $(\mu_0', \tau')$ all outcomes are identically zero in both groups. Hence, these scenarios are completely indistinguishable, and it remains to show that the objective value of (6) is at least the first error term. We defer the complete proofs to the appendix.

## 3 Random Design

The random design assumption is more practical and complicated than the fixed design, as the nearest matching distances of different individuals are typically different without the regular grid structure. In particular, under the random design some covariates have better matching quality than others, in the sense that the minimum pairwise distance between the control and treatment covariates is $\Theta(n^{-2/d})$ while the typical distance is as large as $\Theta(n^{-1/d})$. In this section, we propose a two-stage nearest-neighbor estimator for HTE under the random design, and show that the minimax estimation error exhibits three different behaviors.

### 3.1 Estimator Construction

In the previous section, we construct pseudo-observations at each control covariate using the same treatment grid geometry in the neighborhood of that covariate, thanks to the fixed grid design assumption. However, the local geometries in the random design may differ spatially. For example, for $n$ i.i.d. uniformly distributed random variables on $[0, 1]$, the typical spacing between spatially adjacent values is $\Theta(n^{-1})$, while the minimum spacing is of the order $\Theta(n^{-2})$ with high probability. Therefore, the key insight behind the new estimator construction is to look for the best geometry in each local neighborhood. Specifically, we propose the following two-stage estimator: at the first stage we collect $m_1$ nearest control covariates of the query point, while at the second stage we find the 1-nearest-neighbor treatment covariates for all above control samples and only pick $m_2 \leq m_1$ of them with desirably small nearest neighbor distances. Note that the nearest-neighbor estimator is a better candidate than the kernel-based estimator here, as the nearest-neighbor distance can adapt to different density regimes and result in adaptive bandwidths.

The detailed estimator construction is displayed in Algorithm 2. Note that the first stage is similar in spirit to the kernel estimator, where we simply collect $m_1$ control samples in the small neighborhood and aim to take the (weighted) average. However, the second stage is very different in the sense that it asks to throw away samples with poor covariate matching quality to all treatment covariates and only keeps $m_2$ pairs of samples. Finally, a simple average is taken to the difference of the selected $m_1$ pairs, which is sufficient for smoothness $\beta_\tau \leq 1$. Although Algorithm 2 can be applied to higher smoothness in principle, we will discuss the fundamental challenges of $\beta_\tau > 1$ in Section 5.

---
**Algorithm 2** Estimator Construction under Random Design
---

**Input:** control observations $\{(X_i^0, Y_i^0)\}_{i \in [n]}$, treatment observations $\{(X_i^1, Y_i^1)\}_{i \in [n]}$, dimensionality $d$, number of neighbors $m_1, m_2$ with $m_1 \geq \kappa m_2$, query point $x_0 \in [0, 1]^d$.

**Output:** HTE estimator $\hat\tau(x_0)$ at query point $x_0$.

Find $m_1$ nearest control covariates $\{X_i^0\}_{i \in I_1}$ in the Euclidean distance to $x_0$, with $|I_1| = m_1$;

**for** $i \in I_1$ **do**

    Compute the minimum distance $d_i = \min_{j \in [n]} \|X_i^0 - X_j^1\|_2$ to the treatment covariate;

    Let $j(i) \in [n]$ be any minimizer to the above minimization program.

**end for**

Find the smallest $m_2$ elements of $(d_i)_{i \in I_1}$ and denote the index set by $I_2$, with $|I_2| = m_2$;

Return the final estimate $\hat\tau(x_0) = m_2^{-1} \sum_{i \in I_2} (Y_{j(i)}^1 - Y_i^0)$.

---

The optimal choices of $(m_1, m_2)$ depend on the density ratio $\kappa$ and noise level $\sigma$ as follows:

$$(m_1, m_2) = \widetilde\Theta \begin{cases} \left(\kappa^{\frac{\beta_\mu}{\beta_\tau + \beta_\mu}} n^{\frac{\beta_\tau - \beta_\mu}{\beta_\tau + \beta_\mu}}, 1\right) & \text{if } \sigma \leq \sigma_1 \\ \left(n \cdot (\kappa\sigma^2/n^2)^{\frac{d\beta_\mu}{2\beta_\mu\beta_\tau + d(\beta_\mu+\beta_\tau)}}, (n^2/\kappa)^{\frac{2\beta_\mu\beta_\tau}{2\beta_\mu\beta_\tau + d(\beta_\mu+\beta_\tau)}} \sigma^{\frac{2d(\beta_\mu+\beta_\tau)}{2\beta_\mu\beta_\tau + d(\beta_\mu+\beta_\tau)}}\right) & \text{if } \sigma_1 < \sigma \leq \sigma_2 \\ \left(n^{\frac{2\beta_\tau}{2\beta_\tau + d}} (\sigma^2\kappa)^{\frac{d}{2\beta_\tau + d}}, (n/\kappa)^{\frac{2\beta_\tau}{2\beta_\tau + d}} \sigma^{\frac{2d}{2\beta_\tau + d}}\right) & \text{if } \sigma > \sigma_2 \end{cases}$$

with noise thresholds $(\sigma_1, \sigma_2)$ given by

$$\sigma_1 = (\kappa/n^2)^{1/d(\beta_\mu^{-1} + \beta_\tau^{-1})}, \quad \sigma_2 = n^{(\beta_\tau - \beta_\mu)/(2\beta_\tau) - \beta_\mu/d}/\sqrt{\kappa}.$$

The rationale behind the above choices lies in the following theorem.

**Theorem 5.** *Given Assumptions 1-3 and $\beta_\mu \leq \beta_\tau \leq 1$, fix any integer parameters $(m_1, m_2)$ with $m_1 \geq \kappa m_2$ and $m_2 = \Omega(\log n)$, the following upper bound holds for the estimator $\hat\tau$ in Algorithm 2:*

$$\sup_{\mu_0 \in \mathcal{H}_d(\beta_\mu), \tau \in \mathcal{H}_d(\beta_\tau)} \mathbb{E}_{\mu_0, \tau}[\|\hat\tau - \tau\|_{L_1(g_0)}] \leq \mathsf{polylog}(n) \cdot \left(\left(\frac{\kappa m_2}{n m_1}\right)^{\frac{\beta_\mu}{d}} + \left(\frac{m_1}{n}\right)^{\frac{\beta_\tau}{d}} + \frac{\sigma}{\sqrt{m_2}}\right),$$

*where $\mathsf{polylog}(n)$ hides poly-logarithmic factors in $n$. In particular, if the parameters $(m_1, m_2)$ are chosen as above, the estimator $\hat\tau$ achieves the upper bound of Theorem 2.*

We explain the originations of the three types of errors in Theorem 5. The first error comes from the covariate matching bias, where it is given by the average of the $m_2$ smallest 1-nearest-neighbor distances between control-treatment pairs at the second stage. The second error comes from the bias incurred at the first stage, where the key quantity of interest is the distance between a random query point $x_0 \sim g_0$ to its $m_1$-nearest-neighbor. Finally, the third error comes from the observational noise, where it is expected that the noise magnitude drops off by a factor of $\sqrt{m_2}$ after averaging. Hence, the optimal choice of the parameters $(m_1, m_2)$ aims to balance the above types of errors, with an additional constraint $\kappa m_2 \leq m_1$ to ensure that no single treatment covariate is matched to too many control covariates and therefore lead to a small variance.

The full proof of Theorem 5 is postponed to the appendix, but we remark that an appropriate maximal inequality is the key to handling general densities $g_0, g_1$ which may be close to zero. Specifically, let $f$ be any density function on $[0, 1]^d$, and define the following *minimal function* of $f$ as

$$m[f](x) \triangleq \inf_{0 < r \leq \sqrt{d}} \frac{1}{\mathrm{Vol}_d(B(x; r))} \int_{B(x; r)} f(t) dt, \tag{7}$$

where $B(x; r)$ denotes the ball centered at $x$ with radius $r$, $\mathrm{Vol}_d$ is the $d$-dimensional volume, and $f(t) = 0$ whenever $t \notin [0, 1]^d$. It is clear that $m[f](x) \leq f(x)$ for almost all $x \in [0, 1]^d$, while the following lemma summarizes a key property of $m[f]$ which is very useful in the proof of Theorem 5 and could be of independent interest.

**Lemma 1.** *For every $\lambda > 0$, there exists a constant $C_d > 0$ depending only on $d$ such that*

$$\int_{[0,1]^d} f(x) e^{-\lambda \cdot m[f](x)} dx \leq \begin{cases} \exp(-\lambda/eC_d) & \text{if } \lambda < eC_d, \\ C_d/\lambda & \text{if } \lambda \geq eC_d. \end{cases}$$

## 3.2 Minimax Lower Bound

The following minimax lower bound complements Theorem 5 and shows the near minimax optimality (up to logarithmic factors) of the two-stage estimator in Algorithm 2.

**Theorem 6.** *Assume that $\beta_\mu \leq \beta_\tau \leq 1$. Then under Gaussian noises and appropriate densities $g_0, g_1$ satisfying Assumption 2, the minimax risk under the random design satisfies*

$$R_{n,d,\beta_\mu,\beta_\tau,\sigma}^{\text{random}}(\kappa) \geq \frac{1}{\mathsf{polylog}(n)} \left( \left( \frac{\kappa}{n^2} \right)^{\frac{1}{d(\beta_\mu^{-1}+\beta_\tau^{-1})}} + \left( \frac{\kappa\sigma^2}{n^2} \right)^{\frac{1}{2+d(\beta_\mu^{-1}+\beta_\tau^{-1})}} + \left( \frac{\kappa\sigma^2}{n} \right)^{\frac{\beta_\tau}{2\beta_\tau+d}} \right).$$

The proof of Theorem 6 is very involved and postponed to the appendix. Similar to the fixed design, the last error term comes from the classical nonparametric estimation of $\tau$, and establishing the first error term requires to solve a linear program in the same spirit to (6). However, as a regular grid is no longer available, constructing a near-optimal solution to (6) becomes much more challenging. In addition, proving the second error term requires a hybrid approach of testing multiple hypotheses and constructing an exponential number of feasible solutions to (6). To this end, the following novel representation of the Hölder continuity condition is very useful throughout the lower bound argument.

**Lemma 2.** *Let $\beta \leq 1$, and $\{(x_i, y_i)\}_{i=1}^n$ be fixed covariate-value pairs on $[0,1]^d \times \mathbb{R}$. Then there exists some function $f \in \mathcal{H}_d(\beta)$ (with radius $L$) with $f(x_i) = y_i$ for all $i \in [n]$ if and only if for any $i, j \in [n]$, we have $|y_i - y_j| \leq L\|x_i - x_j\|_2^\beta$.*

## 4 Numerical Experiments

We illustrate the efficacy of the proposed estimator in Algorithm 2 via some numerical experiments. Specifically, we aim to show that the two main ingredients of Algorithm 2, i.e. constructing pseudo-observations based on covariate matching and discarding observations with poor matching quality, are key to improved HTE estimation. We compare our estimator (which we call *selected matching*) with the following three estimators: the *full matching* estimator which never discards samples (i.e. $m_2 = m_1$ always holds in Algorithm 2), the *kNN differencing* and *kernel differencing* estimators which apply separate $k$-NN or kernel estimates to both baselines and then take the difference. Note that the idea of differencing-based estimators were essentially used in [AS18, KSBY19].

The experimental setting is as follows. We choose the parameter values $n \in \{50, 100, \cdots, 1,000\}$, $d, \kappa \in \{1, \cdots, 10\}$, and $\sigma = 2/\sqrt{n}$. Denote by $\phi_{\mu,\sigma^2}(x)$ the pdf of the normal random variable $\mathcal{N}(\mu, \sigma^2)$, the baseline function and the HTE are chosen to be

$$\mu_0(x) = 2\phi_{0.1,0.15^2}(x) + 2.5\phi_{0.4,0.05^2}(x) + 4\phi_{0.8,0.1^2}(x), \quad \tau(x) = \phi_{0,1}(2x - 1)$$

for $d = 1$ and $x \in [0,1]$, and for general dimensions, the variable $x$ is replaced by $\sqrt{d}(\bar{x} - 0.5) + 0.5$. An illustration of $\mu_0$ and $\tau$ for $d = 1$ is plotted in Figure 1, where $\tau$ is smoother and $\mu_0$ is more volatile. For each given $(n, d, \kappa, \sigma)$, we generate $n$ control covariates $X_1^0, \cdots, X_n^0$ following the i.i.d. density $g_0(x) = 2\kappa/(\kappa + 1)$ when $x_1 \in [0, 1/2]$ and $g_0(x) = 2/(\kappa + 1)$ when $x_1 \in (1/2, 1]$. Similarly, the treatment covariates $X_1^1, \cdots, X_n^1$ are i.i.d. generated following the density $g_1(x) = 2 - g_0(x)$, and the responses $Y_i^0, Y_i^1$ are defined in (1) with i.i.d. $\mathcal{N}(0, \sigma^2)$ noises. The algorithm parameters are determined by the optimal bias-variance tradeoffs in theory. Finally, the performance of HTE estimation is measured via the root mean squared error (RMSE) averaged over 100 simulations. The source codes are available at `https://github.com/Mathegineer/NonparametricHTE`.

The experimental results are displayed in Figures 1 and 2, and we have the following observations. First, the differencing based estimators typically perform worse than the matching based estimators, especially at the places where the baseline is more volatile. Therefore, the differencing suffers more from the less smooth baselines. Second, although the *full matching* estimator behaves similarly as *selected matching* when $d = \kappa = 1$, the performance of *full matching* deteriorates significantly if either $d$ or $\kappa$ increases, and *selected matching* is relatively more stable. This observation supports our intuition that the pairs with poor covariate matching quality should be discarded.

## 5 Further Discussions

A major concern of this work is to capture higher order smoothness under the random design, which we remark is very challenging from both sides of achevability and lower bounds. For example,

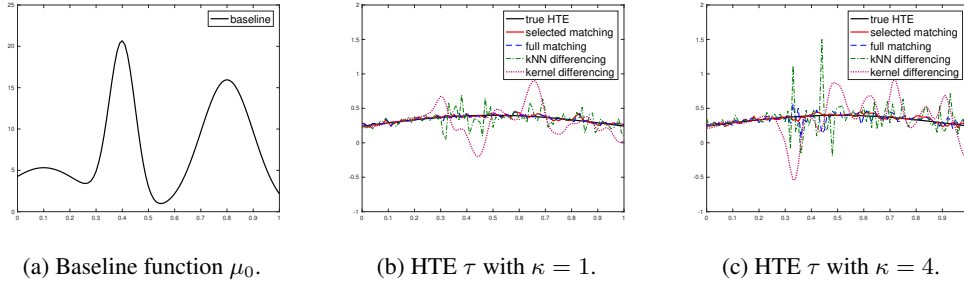

(a) Baseline function $\mu_0$.　　　　(b) HTE $\tau$ with $\kappa = 1$.　　　　(c) HTE $\tau$ with $\kappa = 4$.

Figure 1: The baseline function $\mu_0$, HTE $\tau$, as well as the estimates.

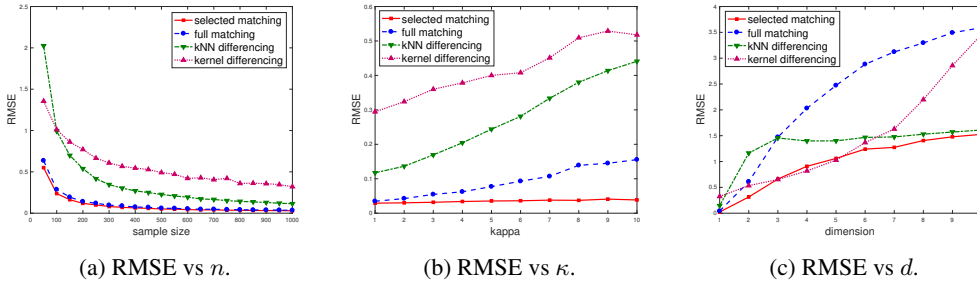

(a) RMSE vs $n$.　　　　(b) RMSE vs $\kappa$.　　　　(c) RMSE vs $d$.

Figure 2: The RMSE as a function of $n$, $\kappa$ or $d$ with default parameters $(n, \kappa, d) = (1000, 1, 1)$.

there are a number of open questions related to capturing higher order smoothness in nonparametric statistics, e.g. the adaptation to small densities [PR$^+$16] despite the known asymptotic equivalence [Nus96], in the estimation of conditional variance [RR14], non-smooth functionals [HJWW17], the analysis of nearest-neighbor methods [JGH18], local goodness-of-fit tests [BW19]. Specializing to our setting, to capture the smoothness $\beta_\mu > 1$ and reduce the matching bias, locally there need to be at least two treatment-control pairs mutually of small distance. However, instead of the $\Theta(n^{-2})$ minimum distance for $n$ uniform random variables in $[0, 1]$, the counterpart for two pairs becomes at least $\Theta(n^{-4/3})$. Hence, we expect that there will be a phase transition from $\beta_\mu \leq 1$ to $\beta_\mu > 1$ in the estimation performance, and it is an outstanding open question to characterize this transition.

As for the lower bound, the main difficulty of the extension to $\beta \geq 1$ lies in Lemma 2. To the best of our knowledge, there is no simple and efficient criterion for the existence of a Hölder $\beta$-smooth function with specified values on general prescribed points. To illustrate the difficulty, assume that $\beta$ is an integer and consider the following natural generalization of the condition in Lemma 2: for any $i_0, i_1, \cdots, i_\beta \in [n]$, it holds that $\beta! \cdot |f[x_{i_0}, x_{i_1}, \cdots, x_{i_\beta}]| \leq L$, where $f[x_0, x_1, \cdots, x_m]$ is the divided difference of $f$ on points $x_0, x_1, \cdots, x_m$ [dB05]. Standard approximation theory shows that this condition is necessary, and it reduces to Lemma 2 for $\beta = 1$. However, it is not sufficient when $\beta \geq 2$. For a counterexample, the value specification $f(-3) = f(-1) = 1, f(1) = f(3) = 0$ satisfies the divided difference condition with $\beta = 2, L = 1/4$, but no choice of $f(0)$ satisfies both $|f[-3, -1, 0]| \leq 1/8$ and $|f[0, 1, 3]| \leq 1/8$.

We also point out some other extensions. First, the current estimators rely on the knowledge of many parameters which are unlikely to be known in practice. For the smoothness parameters and the density ratio parameter, their impact factors through the bias and it is possible to apply Lepski's trick [LMS97] or data-driven parameters [FG95] to achieve full or partial adaptation. For other parameters such as the noise level, pre-estimation procedures such as [DJKP95] are expected to be helpful. Second, in our work, we study the minimax risk after conditioning on both random realizations of the covariates and group assignments, meaning that the worst-case HTE can depend on both the above realizations. In contrast, [NW17] only conditions on the random covariates and considers an expected risk taken with respect to the randomness in the group assignments, where the worst-case HTE cannot depend on the group assignments. The minimax risk in the latter setting remains unknown, and it is an outstanding question to compare the minimax rates in these settings.

## Broader Impact

This work mainly provides theoretical tools and bounds for the HTE estimation in causal inference, as well as potentially useful practical insights such as the two-stage nearest-neighbors and throwing away observations with poor covariate matching quality. This special form of nonparametric estimation problems could be a useful addition to the literature in nonparametric statistics, and theorists and practitioners working on causal inference may potentially benefit from this work.

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
