[Supplementary Material]

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

# A    Auxiliary Lemmas

**Lemma 3.** *Fix any $t \in \mathbb{N}$, $m \geq 1$ and $0 \leq \Delta \leq 1/(2m)$. For any $i \in [m]$, let $x_0, \cdots, x_t$ be the $(t+1)$ nearest neighbors of $(i-1)/m$ in $\{\Delta, 1/m+\Delta, \cdots, 1-1/m+\Delta\}$ with increasing distance, and the weights $w_0, \cdots, w_t$ be the solution to*

$$\sum_{j=0}^{t} w_j = 1, \qquad \sum_{j=0}^{t} w_j (x - x_j)^{\ell} = 0, \qquad \ell = 1, \cdots, t.$$

*Then there exists a constant $C > 0$ depending only on $t$ such that*

$$|w_0| \leq C, \qquad |w_j| \leq C \cdot m\Delta, \qquad j = 1, 2, \cdots, t.$$

**Lemma 4** (Chernoff bound, Theorem 5.4 of [MU05]). *For $X \sim \mathsf{B}(n, \lambda/n)$ and any $\delta \in (0, 1]$, we have*

$$\mathbb{P}(X \geq (1+\delta)\lambda) \leq \left(\frac{e^{\delta}}{(1+\delta)^{1+\delta}}\right)^{\lambda} \leq \exp\left(-\frac{\delta^2 \lambda}{3}\right),$$

$$\mathbb{P}(X \leq (1-\delta)\lambda) \leq \left(\frac{e^{-\delta}}{(1-\delta)^{1-\delta}}\right)^{\lambda} \leq \exp\left(-\frac{\delta^2 \lambda}{2}\right).$$

**Lemma 5** (Generalized Hardy–Littlewood maximal inequality, a slight variant of Lemma 3 of [JGH18]). *Let $\mu_1, \mu_2$ be two Borel measures that are finite on the bounded Borel sets of $\mathbb{R}^d$. Then, for all $t > 0$ and any bounded Borel set $A \subseteq \mathbb{R}^d$,*

$$\mu_1\left(\left\{x \in A : \sup_{\rho > 0 : B(x;\rho) \subseteq A} \left(\frac{\mu_2(B(x;\rho))}{\mu_1(B(x;\rho))}\right) \geq t\right\}\right) \leq \frac{C_d}{t}\mu_2(A).$$

*Here $C_d > 0$ is a constant that depends only on the dimension $d$.*

# B    Proof of Main Theorems

Throughout the proofs, we introduce the following notations: in addition to the asymptotic notation $a_n = \Theta(b_n)$, we also write $a_n = O(b_n)$ if $\limsup_{n \to \infty} a_n/b_n < \infty$, and $a_n = \Omega(b_n)$ if $b_n = O(a_n)$. Similarly, the notations $\widetilde{O}(\cdot), \widetilde{\Omega}(\cdot)$ denote the respective meanings within logarithmic factors in $n$.

## B.1    Proof of Theorem 1

The minimax risk of HTE estimation under the fixed design is a direct consequence of Theorems 3 and 4.

### B.1.1    Proof of Theorem 3

By the definition of the pseudo-observation $\hat{Y}^1(x)$ and the potential outcome model (1), we have

$$\hat{Y}^1(X_i^0) - Y_0^i = \tau(X_i^0) + b_{\mu}(X_i^0) + \tilde{\varepsilon}_i, \tag{8}$$

with the new errors $\tilde{\varepsilon}_i$ defined as (with $t = \lfloor \beta_{\mu} \rfloor + 1$)

$$\tilde{\varepsilon}_i = \sum_{j \in [t]^d} w_j \cdot \varepsilon_j^1 - \varepsilon_i^0,$$

and the matching bias $b_{\mu}(X_i^0)$ is

$$b_{\mu}(X_i^0) = \sum_{j \in [t]^d} w_j \cdot (\mu_0(x_j) + \tau(x_j)) - (\mu_0(X_i^0) + \tau(X_i^0)),$$

where $w_j, x_j$ are the weights and the treatment covariates used to obtain pseudo-observations at $X_i^0$. By the smoothness property of $\mu_1 = \mu_0 + \tau$ in Assumption 1, the Taylor expansion around $X_i^0$ gives

$$\mu_1(x_j) = \sum_{\beta=0}^{\lfloor \beta_\mu \rfloor - 1} \sum_{\sum_{i=1}^d \beta_i = \beta} \frac{\partial^\beta \mu_1(X_i^0)}{\partial x_1^{\beta_1} \cdots \partial x_d^{\beta_d}} \prod_{k=1}^d \frac{(x_{j,k} - X_{i,k}^0)^{\beta_k}}{\beta_k!}$$

$$+ \sum_{\sum_{i=1}^d \beta_i = \lfloor \beta_\mu \rfloor} \frac{\partial^\beta \mu_1(\xi_j)}{\partial x_1^{\beta_1} \cdots \partial x_d^{\beta_d}} \prod_{k=1}^d \frac{(x_{j,k} - X_{i,k}^0)^{\beta_k}}{\beta_k!},$$

where $\xi_j \in \mathbb{R}^d$ is some point on the segment connecting $X_i^0$ and $x_j$. Hence, by the smoothness assumption of $\mu_1$, it holds that

$$\left| \mu_1(x_j) - \sum_{\beta=0}^{\lfloor \beta_\mu \rfloor} \sum_{\sum_{i=1}^d \beta_i = \beta} \frac{\partial^\beta \mu_1(X_i^0)}{\partial x_1^{\beta_1} \cdots \partial x_d^{\beta_d}} \prod_{k=1}^d \frac{(x_{j,k} - X_{i,k}^0)^{\beta_k}}{\beta_k!} \right| \le C \sum_{(\beta_1, \cdots, \beta_d) \in \Gamma} \prod_{k=1}^d |x_{j,k} - X_{i,k}^0|^{\beta_k},$$

where $C > 0$ is an absolute constant, and

$$\Gamma = \left\{ (\beta_1, \cdots, \beta_d) : \beta_k \ge 0, \sum_{k=1}^d \beta_k = \beta_\mu, \text{at most one of } \beta_k \text{ is non-integer} \right\}$$

is a finite set of indices. Now by the choice of each $w_j$ in (3), we have

$$|b_\mu(X_i^0)| \le C \sum_{(\beta_1, \cdots, \beta_d) \in \Gamma} \prod_{k=1}^d \left( \sum_{j=1}^t |w_{j,k}| \cdot |x_{j,k} - X_{i,k}^0|^{\beta_k} \right). \tag{9}$$

Consider the term $|w_{j,k}| \cdot |x_{j,k} - X_{i,k}^0|^{\beta_k}$ for each index $j \in [t]$ and $k \in [d]$. If $x_{j,k}$ is the nearest neighbor of $X_{i,k}^0$ in the one-dimensional grid $\{\Delta_k, 1/m + \Delta_k, \cdots, 1 - 1/m + \Delta_k\}$, then $|x_{j,k} - X_{i,k}^0| = \Delta_k$ and $|w_{j,k}| = O(1)$ by Lemma 3. Otherwise, we have $|x_{j,k} - X_{i,k}^0| \le t/m$ and $|w_{j,k}| = O(m\Delta_k)$ again by Lemma 3. Therefore, the following inequality always holds:

$$|w_{j,k}| \cdot |x_{j,k} - X_{i,k}^0|^{\beta_k} = O\left( m^{-\beta_k}(m\Delta_k)^{\beta_k \wedge 1} \right) = O\left( m^{-\beta_k}(m\|\Delta\|_\infty)^{\beta_k \wedge 1} \right).$$

Consequently, for any $i \in [n]$ it holds that

$$|b_\mu(X_i^0)| = O\left( Ct^d \cdot |\Gamma| \max_{(\beta_1, \cdots, \beta_d) \in \Gamma} \left( \prod_{k=1}^d m^{-\beta_k}(m\|\Delta\|_\infty)^{\beta_k \wedge 1} \right) \right) = O\left( m^{-\beta_\mu}(m\|\Delta\|_\infty)^{\beta_\mu \wedge 1} \right),$$

which gives the claimed matching bias upper bound in the error decomposition of Theorem 3.

The rest of the proof follows from the classic analysis of the Nadaraya-Watson estimator applied to (8) (see, e.g. [Tsy09]), where the only difference is that the errors $\tilde{\varepsilon}_i$ are weakly dependent instead of being mutually independent, i.e. each $\tilde{\varepsilon}_i$ depends only on finitely many $\tilde{\varepsilon}_j$'s.

### B.1.2 Proof of Theorem 4

To prove the lower bound $\Omega((\sigma^2/n)^{\beta_\tau/(2\beta_\tau + d)})$, consider $\mu_0 \equiv 0$. In this scenario, the outcomes of the control group are completely non-informative for estimating $\tau$, and the treatment outcome model (1) is precisely the nonparametric regression model for the $\beta_\tau$-Hölder smooth function $\tau$. Then by the standard minimax lower bound for nonparametric regression (see, e.g. [Tsy09]), the lower bound $\Omega((\sigma^2/n)^{\beta_\tau/(2\beta_\tau + d)})$ is immediate.

Next we prove the lower bound $\Omega(n^{-\beta_\mu/d}(n^{1/d}\|\Delta\|_\infty)^{\beta_\mu \wedge 1})$, which corresponds to the matching bias. We first construct a feasible solution $(\mu_0, \tau)$ to the optimization problem (6). Without loss of generality we assume that $\Delta_1 = \|\Delta\|_\infty$. Construct the following function $g$ on $[0,1]^d$: for $x = (x_1, \cdots, x_d) \in [0,1]^d$,

$$g(x) = c \cdot [x_1(1 - x_1)]^{\beta_\mu \wedge 1}$$

with some small constant $c > 0$. We claim that $g$ is $\beta$-Hölder smooth on $[0, 1]^d$ given a sufficiently small $c > 0$. In fact, if $\beta_\mu \le 1$, for all $x_1, x_1' \ge 0$ simple algebra gives $|(x_1')^{\beta_\mu} - x_1^{\beta_\mu}| \le |x_1' - x_1|^{\beta_\mu}$, and therefore

$$
\begin{aligned}
|g(x) - g(x')| &= c \cdot \left| [x_1(1 - x_1)]^{\beta_\mu} - [x_1'(1 - x_1')]^{\beta_\mu} \right| \\
&\le c \cdot \left( (1 - x_1)^{\beta_\mu} |x_1^{\beta_\mu} - (x_1')^{\beta_\mu}| + (x_1')^{\beta_\mu} |(1 - x_1)^{\beta_\mu} - (1 - x_1')^{\beta_\mu}| \right) \\
&\le c \cdot \left( 1 \cdot |x_1 - x_1'|^{\beta_\mu} + 1 \cdot |x_1 - x_1'|^{\beta_\mu} \right) \\
&= 2c|x_1 - x_1'|^{\beta_\mu} \le 2c\|x - x'\|_2^{\beta_\mu}.
\end{aligned}
$$

If $\beta_\mu > 1$, the function $g(x) = cx_1(1 - x_1)$ is a smooth function in $x$ and therefore $\beta_\mu$-Hölder smooth as well.

Recall that $(0, 1]^d = \cup_{i=1}^n (X_i^0 + (0, m^{-1}]^d)$ under the fixed design. Consider the following construction of $\mu_0$:

$$
\mu_0(x) = h(x) \cdot \frac{1}{m^{\beta_\mu}} \sum_{i=1}^n g\left( m(x - X_i^0) \right) \mathbb{1}(m(x - X_i^1) \in (0, 1]^d),
$$

where the function $g$ is defined above, and $h$ is an arbitrary smooth function on $\mathbb{R}^d$ supported on $[0, 1]^d$ with $h(x) \equiv 1$ for $x \in [1/4, 3/4]^d$. In other words, the baseline function $\mu_0$ is a dilation of the reference function $g$ with proper scaling to preserve $\beta_\mu$-smoothness, and the function $h$ preserves the value of $\mu_0$ in the center of $[0, 1]^d$ while helps $\mu_0$ connect to zero smoothly at the boundary of $[0, 1]^d$. Since multiplying a bounded smooth function does not decrease the smoothness parameter, it is straightforward to verify that $\mu_0 \in \mathcal{H}_d(\beta_\mu)$ fulfills Assumption 1.

We also construct the HTE $\tau$ as $\tau(x) = -m^{-\beta_\mu} g(m\Delta) \cdot h(x)$, where $h$ is the same smooth function used in the definition of $\mu_0$. By the smoothness of $h$, it is clear that $\tau \in \mathcal{H}_d(\beta_\tau)$ for any $\beta_\tau > 0$. Moreover, the pair $(\mu_0, \tau)$ is a feasible solution to (6): the smoothness conditions have already been verified, and

$$
\mu_0(X_i^0) = h(X_i^0) \cdot m^{-\beta_\mu} g(0) = 0,
$$
$$
\mu_0(X_i^1) + \tau(X_i^1) = h(X_i^1) \cdot m^{-\beta_\mu} g(m\Delta) - m^{-\beta_\mu} g(m\Delta) \cdot h(X_i^1) = 0.
$$

Also, the construction of $g$ ensures that $|g(m\Delta)| = \Omega((m\|\Delta\|_\infty)^{\beta_\mu \wedge 1})$, and the fact $h(x) \equiv 1$ for $x \in [1/4, 3/4]^d$ gives $\|h\|_1 = \Omega(1)$. Consequently, we have

$$
\|\tau\|_1 = m^{-\beta_\mu} |g(m\Delta)| \cdot \|h\|_1 = \Omega\left( n^{-\frac{\beta_\mu}{d}} (n^{\frac{1}{d}} \|\Delta\|_\infty)^{\beta_\mu \wedge 1} \right).
$$

Finally, note that the distributions of the outcomes (1) in two groups are the same under the above construction of $(\mu_0, \tau)$ and the naïve construction $(\mu_0' \equiv 0, \tau' \equiv 0)$, the standard two-point method yields to a minimax lower bound $\|\tau - \tau'\|_2$, which is exactly the desired matching bias.

## B.2 Proof of Theorem 2

The minimax risk of HTE estimation under the random design is a direct consequence of Theorems 5 and 6.

### B.2.1 Proof of Theorem 5

For each $x_0 \in [0, 1]^d$, the construction of the two-stage nearest-neighbor based estimator $\hat{\tau}(x_0)$ in Algorithm 2 gives

$$
\hat{\tau}(x_0) - \tau(x_0) = \frac{1}{m_2} \sum_{i \in I_2} (\mu_0(X_{j(i)}^1) - \mu_0(X_i^0)) + \frac{1}{m_2} \sum_{i \in I_2} (\tau(X_{j(i)}^1) - \tau(x_0)) + \frac{1}{m_2} \sum_{i \in I_2} (\varepsilon_{j(i)}^1 - \varepsilon_i^0)
$$
$$
\triangleq b_\mu(x_0) + b_\tau(x_0) + s(x_0),
$$

where the above terms correspond to the matching bias incurred by $\mu$, estimation bias incurred by $\tau$, and the stochastic error incurred by the noises, respectively. As the final estimation error is measured

under $L_1(g_0)$, it suffices to upper bound the quantites $\mathbb{E}|b_\mu(X)|$, $\mathbb{E}|b_\tau(X)|$, and $\mathbb{E}|s(X)|$ for a fresh test covariate $X \sim g_0$, respectively.

We first upper bound the quantity $\mathbb{E}|b_\mu(X)|$. As in the description of the Algorithm 2, for $i \in I_1$, let $d_i = \min_{j \in [n]} \|X_i^0 - X_j^1\|_2$ be the minimum Euclidean distance between the control covariate $X_i^0$ and all possible treatment covariates, and $D_1$ be the random variable representing the $m_2$-th smallest value of $(d_i)_{i \in I_1}$. Then by the smoothness condition of $\mu_0$, it is clear that

$$\mathbb{E}|b_\mu(X)| \lesssim \mathbb{E}[D_1^{\beta_\mu}] \le (\mathbb{E}[D_1])^{\beta_\mu}. \tag{10}$$

Hence, it remains to upper bound the expectation $\mathbb{E}[D_1]$. First, for any $x \in [0,1]^d$, $j \in [n]$ and $t \in (0, \sqrt{d})$, we have

$$\mathbb{P}\left(\|X_j^1 - x\|_2 \ge t\right) = \int_{y \in [0,1]^d : y \notin B(x;t)} g_1(y)dy \le 1 - \gamma_d t^d \cdot m[g_1](x),$$

by the definition of the minimal function defined in (7), where $\gamma_d$ is the volume of the $d$-dimensional unit ball. Consequently, for each $i \in I_1$ and $0 < t \lesssim n^{-1/d}$, we have

$$\begin{aligned}
\mathbb{P}(d_i \ge t) &= \mathbb{E}\left[\mathbb{P}\left(\|X_1^1 - X_i^0\|_2 \ge t \mid X_i^0\right)^n\right] \\
&\le \mathbb{E}\left[\left(1 - \gamma_d t^d \cdot m[g_1](X_i^0)\right)^n\right] \\
&\overset{(a)}{\le} \mathbb{E}\left[\exp\left(-n\gamma_d t^d \cdot m[g_1](X_i^0)\right)\right] \\
&= \int_{[0,1]^d} g_0(x) \exp\left(-n\gamma_d t^d m[g_1](x)\right) dx \\
&\overset{(b)}{\le} \int_{[0,1]^d} g_0(x) \exp\left(-\frac{n\gamma_d t^d}{\kappa} m[g_0](x)\right) dx \\
&\overset{(c)}{\le} \exp\left(-\frac{c_d n t^d}{\kappa}\right),
\end{aligned}$$

where (a) is due to the elementary inequality $1 - x \le \exp(-x)$, (b) follows from the upper bound on the density ratio, and (c) is due to Lemma 1 and the assumption $t \lesssim n^{-1/d}$, with some absolute constant $c_d > 0$. Now by the mutual independence of control and treatment covariates, the random variable $D_1$ is simply the $m_2$-th order statistic of $(d_i)_{i \in I_1}$, and therefore

$$\begin{aligned}
\mathbb{P}(D_1 \ge t) &= \mathbb{P}\left(\mathsf{B}\left(m_1, \mathbb{P}(d_1 \ge t)\right) \ge m_1 - m_2 + 1\right) \\
&\le \mathbb{P}\left(\mathsf{B}\left(m_1, \exp(-c_d n t^d/\kappa)\right) \ge m_1 - m_2 + 1\right) \\
&= \mathbb{P}\left(\mathsf{B}\left(m_1, 1 - \exp(-c_d n t^d/\kappa)\right) < m_2\right).
\end{aligned}$$

Hence, for $t \ge C_d(\kappa m_2/nm_1)^{1/d}$ with a large enough constant $C_d > 0$, we have

$$1 - \exp\left(-c_d n t^d/\kappa\right) \le \frac{m_2}{2m_1},$$

and therefore Chernoff bound (Lemma 4) gives the upper bound $\mathbb{P}(D_1 \ge t) = \exp(-\Omega(m_2))$, which is smaller than $O(n^{-1})$ by the assumption that $m_2$ is at least logarithmic in $n$. Consequently,

$$\mathbb{E}[D_1] = \int_0^{\sqrt{d}} \mathbb{P}(D_1 \ge t)dt \le C_d \left(\frac{\kappa m_2}{nm_1}\right)^{\frac{1}{d}} + \sqrt{d} \cdot \frac{1}{n} \lesssim \left(\frac{\kappa m_2}{nm_1}\right)^{\frac{1}{d}},$$

and consequently (10) leads to

$$\mathbb{E}|b_\mu(X)| \lesssim \left(\frac{\kappa m_2}{nm_1}\right)^{\frac{\beta_\mu}{d}}. \tag{11}$$

We then upper bound the quantity $\mathbb{E}|b_\tau(X)|$. Let $D_2$ be the non-negative random variable representing the Euclidean distance between the query point $X$ and its $m_1$-nearest neighbor in the control group,

then standard nearest neighbor analysis (see, e.g. [BD15, Theorem 2.4]) gives $\mathbb{E}[D_2] \lesssim (m_1/n)^{1/d}$. Consequently,

$$
\begin{aligned}
\mathbb{E}|b_\tau(X)| &\leq \frac{1}{m_2} \sum_{i \in I_2} \mathbb{E}\left( |\tau(X^1_{j(i)}) - \tau(X^0_i)| + |\tau(X^0_i) - \tau(X)| \right) \\
&\lesssim \mathbb{E}[D_1^{\beta_\tau}] + \mathbb{E}[D_2^{\beta_\tau}] \\
&\leq (\mathbb{E}[D_1])^{\beta_\tau} + (\mathbb{E}[D_2])^{\beta_\tau} \\
&\lesssim \left( \frac{\kappa m_2}{n m_1} \right)^{\frac{\beta_\tau}{d}} + \left( \frac{m_1}{n} \right)^{\frac{\beta_\tau}{d}} \\
&\lesssim \left( \frac{m_1}{n} \right)^{\frac{\beta_\tau}{d}} ,
\end{aligned}
\tag{12}
$$

where the last inequality follows from the assumption that $\kappa m_2 \leq m_1$.

Finally we deal with the stochastic error $\mathbb{E}|s(X)|$. Clearly, due to the mutual independence of the noises in Assumption 3, we have

$$
\mathbb{E}[s(X)^2] \lesssim \frac{\sigma^2}{m_2^2} \mathbb{E}\left[ \sum_{j \in [n]} S_j^2 \right],
$$

where $S_j = \sum_{i \in I_2} \mathbb{1}(i \to j)$ is the total number of times $X^1_j$ is chosen to be the nearest neighbor, and $i \to j$ denotes that $X^1_j$ is the nearest neighbor of $X^0_i$ among the treatment covariates. Consequently,

$$
\begin{aligned}
\mathbb{E}\left[ \sum_{j \in [n]} S_j^2 \right] &= \mathbb{E}\left[ \sum_{i \in I_2} \sum_{i' \in I_2} \sum_{j \in [n]} \mathbb{1}(i \to j)\mathbb{1}(i' \to j) \right] \\
&= \mathbb{E}\left[ \sum_{i \in I_2} \sum_{j \in [n]} \mathbb{1}(i \to j) \right] + \mathbb{E}\left[ \sum_{i \neq i' \in I_2} \sum_{j \in [n]} \mathbb{1}(i \to j)\mathbb{1}(i' \to j) \right] \\
&= m_2 + \mathbb{E}\left[ \sum_{i \neq i' \in I_2} \sum_{j \in [n]} \mathbb{1}(i \to j)\mathbb{1}(i' \to j) \right],
\end{aligned}
\tag{13}
$$

where the last identity follows from $\sum_{j \in [n]} \mathbb{1}(i \to j) = 1$. To deal with the second term, first assume that the control and treatment covariates have the same distribution. Then

$$
\mathbb{E}\left[ \sum_{i \neq i' \in I_2} \sum_{j \in [n]} \mathbb{1}(i \to j)\mathbb{1}(i' \to j) \right] \leq \mathbb{E}\left[ \sum_{i \in I_2} \sum_{i' \in [n], i' \neq i} \sum_{j \in [n]} \mathbb{1}(i \to j)\mathbb{1}(i' \to j) \right],
$$

and we only need to upper bound each quantity $\mathbb{E}[\mathbb{1}(i \to j)\mathbb{1}(i' \to j)]$ for $i \in I_2, i' \in [n]\backslash\{i\}$, and $j \in [n]$. By the i.i.d. assumption, the covariates $X^1_1, \cdots, X^1_n, X^0_{i'}$ are i.i.d. conditioning on $X^0_i$. This observation motivates us to consider the following problem: let $Y_0 = x \in [0,1]^d$ be any fixed point, and $Y_1, \cdots, Y_{n+1}$ be i.i.d. distributed as $X^1_1$. For $i \in [n+1] \cup \{0\}$ and $j \in [n+1]$ with $i \neq j$, let $d(i,j) \in [n]$ be the integer $d$ such that $Y_j$ is the $d$-th nearest neighbor of $Y_i$ among $\{Y_1, \cdots, Y_{i-1}, Y_{i+1}, \cdots, Y_{n+1}\}$. Consequently, identifying $X^0_i, X^0_{i'}, X^1_j$ as $Y_0, Y_{n+1}$, and $Y_1$ respectively, the i.i.d. assumption gives

$$
\mathbb{E}\left[ \mathbb{1}(i \to j)\mathbb{1}(i' \to j) \mid X^0_i = x \right] \leq \mathbb{E}[\mathbb{1}(d(0,1) \leq 2)\mathbb{1}(d(n+1,1) = 1)].
$$

To upper bound the RHS, note that the exchangeability of $(X_1, \cdots, X_{n+1})$ yields

$$\mathbb{E}[\mathbb{1}(d(0,1) \le 2)\mathbb{1}(d(n+1,1)=1)] = \frac{1}{n(n+1)} \sum_{i \ne j \in [n+1]} \mathbb{E}[\mathbb{1}(d(0,i) \le 2)\mathbb{1}(d(j,i)=1)]$$

$$= \frac{1}{n(n+1)}\mathbb{E}\left[ \sum_{i=1}^{n+1} \mathbb{1}(d(0,i) \le 2) \sum_{j \ne i} \mathbb{1}(d(j,i)=1) \right]$$

$$\overset{(a)}{\le} \frac{1}{n(n+1)}\mathbb{E}\left[ \sum_{i=1}^{n+1} \mathbb{1}(d(0,i) \le 2) \cdot c_d \right]$$

$$\overset{(b)}{=} \frac{2c_d}{n(n+1)},$$

where (a) follows from the deterministic inequality $\sum_{j \ne i} \mathbb{1}(d(j,i)=1) \le c_d$ in [GKOV17, Lemma C.1], i.e. each $X_j$ can only be the nearest neighbor of at most $c_d$ points, with constant $c_d$ depending only on $d$. The identity (b) trivially follows from the definition of nearest-neighbors and $d(0,i)$. Consequently, combining the above displays and (13), we arrive at

$$\mathbb{E}\left[ \sum_{j \in [n]} S_j^2 \right] \le (1 + 2c_d)m_2,$$

and therefore

$$\mathbb{E}[|s(X)|] \le \sqrt{\mathbb{E}[s(X)^2]} \lesssim \frac{\sigma}{\sqrt{m_2}}. \tag{14}$$

In the general case, as $I_2$ is essentially a uniformly random subset of $I_1$, we have

$$\mathbb{E}\left[ \sum_{i \ne i' \in I_2} \sum_{j \in [n]} \mathbb{1}(i \to j)\mathbb{1}(i' \to j) \right] \lesssim \left( \frac{m_2}{m_1} \right)^2 \mathbb{E}\left[ \sum_{i \ne i' \in I_1} \sum_{j \in [n]} \mathbb{1}(i \to j)\mathbb{1}(i' \to j) \right]$$

$$\le \left( \frac{m_2}{m_1} \right)^2 \mathbb{E}\left[ \sum_{i \in I_1} \sum_{i' \in [n], i' \ne i} \sum_{j \in [n]} \mathbb{1}(i \to j)\mathbb{1}(i' \to j) \right].$$

Moreover, by the likelihood ratio assumption, a simple change of measure also gives

$$\mathbb{E}\left[ \mathbb{1}(i \to j)\mathbb{1}(i' \to j) \mid X_i^0 = x \right] \le \kappa \cdot \mathbb{E}[\mathbb{1}(d(0,1) \le 2)\mathbb{1}(d(n+1,1)=1)].$$

Consequently, as $m_1 \ge \kappa m_2$, we again arrive at the same upper bound (14), and the claimed Theorem 5 follows from (11), (12), and (14).

### B.2.2  Proof of Theorem 6

We first show that to prove the lower bound of the minimax rate, it suffices to consider the case where both $g_0$ and $g_1$ are uniform over $[0,1]^d$, and there are $n$ and $m = n/\kappa$ (assumed to be an integer as $\kappa \le n$) observations for the control and treatment groups, respectively. To see this, note that we may choose $g_0$ to be the uniform distribution on $[0,1]^d$, and $g_1$ be $1/\kappa$ in a smaller cube and $\kappa$ otherwise. Note that the smaller cube has the volume $\kappa/(\kappa+1)$, which is at least $\Omega(1)$. Besides, the conditional distribution of $g_1$ restricted to the smaller cube is again uniform, and with high probability there are $O(n)$ control observations and $O(n/\kappa)$ treatment observations in the smaller cube. Consequently, restricting everything including the minimax risk to the smaller cube and using a proper scaling of constant factor, the desired reduction holds for the purpose of proving the lower bound. Furthermore, in this case the $L_1(g_0)$ norm coincides with the usual $L_1$ norm.

As in the proof of Theorem 4, after choosing $\mu \equiv 0$, the lower bound $\Omega((\sigma^2/m)^{\beta_\tau/(2\beta_\tau+d)})$ follows directly from the known results in nonparametric estimation. Hence, using the relationship $m = n/\kappa$, the last error term of Theorem 6 is established. For the other two errors, we will need to construct appropriate functions in $\mathcal{H}_d(\beta_\mu)$ and $\mathcal{H}_d(\beta_\tau)$, and by multiplying a smooth function supported on

$[0,1]^d$ (as in the proof of Theorem 4) we will not consider the boundary effects in the construction below.

First we prove the lower bound $\widetilde{\Omega}((\kappa/n^2)^{1/d(\beta_\mu^{-1}+\beta_\tau^{-1})})$ via providing another feasible solution to the optimization problem (6). Specifically, let $\mu_0, \tau$ be the Hölder smooth functions with their respective smoothness parameters $\beta_\mu, \beta_\tau$ assured by Lemma 2 based on the following value specifications:

$$\mu_0(X_i^0) = 0, \qquad \mu_0(X_i^1) = -\tau(X_i^1) = c \cdot \min_{j \in [m]} \left( \|X_i^1 - X_j^1\|_2^{\beta_\tau} + \min_{k \in [n]} \|X_j^1 - X_k^0\|_2^{\beta_\mu} \right),$$

for all $i \in [m]$, where $c > 0$ is a small constant. However, before applying Lemma 2, we need to show that the conditions of Lemma 2 hold for the above value specifications.

For the HTE function $\tau$, using the Lipschitz property of the minimum $|\min_k a_k - \min_k b_k| \leq \max_k |a_k - b_k|$, we have

$$|\tau(X_i^1) - \tau(X_{i'}^1)|$$
$$\leq c \cdot \max_{j \in [m]} \left| \|X_i^1 - X_j^1\|_2^{\beta_\tau} + \min_{k \in [n]} \|X_j^1 - X_k^0\|_2^{\beta_\mu} - \left( \|X_{i'}^1 - X_j^1\|_2^{\beta_\tau} + \min_{k \in [n]} \|X_j^1 - X_k^0\|_2^{\beta_\mu} \right) \right|$$
$$= c \cdot \max_{j \in [m]} \left| \|X_i^1 - X_j^1\|_2^{\beta_\tau} - \|X_{i'}^1 - X_j^1\|_2^{\beta_\tau} \right|$$
$$\leq c \cdot \|X_i^1 - X_{i'}^1\|_2^{\beta_\tau},$$

where the last step follows from $||a|^{\beta_\tau} - |b|^{\beta_\tau}| \leq |a - b|^{\beta_\tau}$ for all $\beta_\tau \in [0,1]$. Hence, for constant $c > 0$ sufficiently small, the condition of Lemma 2 holds for $\tau$.

For the baseline function $\mu_0$, using the same analysis we have $|\mu_0(X_i^1) - \mu_0(X_{i'}^1)| = O(\|X_i^1 - X_{i'}^1\|_2^{\beta_\tau}) = O(\|X_i^1 - X_{i'}^1\|_2^{\beta_\mu})$ for the treatment-treatment pairs. However, for $\mu_0$ we also need to verify the condition of Lemma 2 for control-control and treatment-control pairs. The the control-control pairs are easy to verify: $|\mu_0(X_i^0) - \mu_0(X_{i'}^0)| = 0$ always holds. As for the treatment-control pairs, we have

$$|\mu_0(X_i^1) - \mu_1(X_{i'}^0)| = |\mu_0(X_i^1)|$$
$$= c \cdot \min_{j \in [m]} \left( \|X_i^1 - X_j^1\|_2^{\beta_\tau} + \min_{k \in [n]} \|X_j^1 - X_k^0\|_2^{\beta_\mu} \right)$$
$$\leq c \cdot \min_{k \in [n]} \|X_i^1 - X_k^0\|_2^{\beta_\mu}$$
$$\leq c \cdot \|X_i^1 - X_{i'}^0\|_2^{\beta_\mu},$$

as desired. Hence, both functions $\mu_0$ and $\tau$ can be extended to Hölder smooth functions on $[0,1]^d$ with desired smoothness parameters, and $(\mu_0, \tau)$ is a feasible solution to the optimization problem (6).

Next we provide a lower bound of $\|\tau\|_1$ for the above construction of $\tau$. Choose a bandwidth

$$h_1 = \widetilde{\Theta}\left( (mn)^{-\frac{\beta_\mu}{d(\beta_\tau + \beta_\mu)}} \right)$$

and define

$$I = \left\{ i \in [m] : \min_{j \in [n]} \|X_i^1 - X_j^0\|_2 \leq \frac{1}{h_1} \left( \frac{1}{mn \log n} \right)^{\frac{1}{d}} \right\}.$$

The next lemma shows that the cardinality of $I$ is upper bounded by $1/(3h^d)$ with high probability.

**Lemma 6.** *If $h_1 \leq c(\log n)^{-2/d}$ with constant $c > 0$ small enough, then with probability at least $1 - n^{-3}$ we have $|I| \leq 1/(3h^d)$ for sufficiently large $n$.*

*Proof.* As the control covariates follow the uniform distribution, for any $x \in [0,1]^d$ we have

$$\mathbb{P}\left( \|x - X_j^0\|_2 \geq \frac{1}{h_1(mn \log n)^{1/d}} \right) \geq 1 - v_d \left( \frac{1}{h_1(mn \log n)^{1/d}} \right)^d = 1 - \frac{v_d}{mn h_1^d \log n},$$

where $v_d$ is the volume of the unit ball in $d$ dimensions. Hence, for each individual $i \in [m]$ in the treatment group, we have

$$\mathbb{P}(E_i) \triangleq \mathbb{P}\left(\min_{j \in [n]} \|X_i^1 - X_j^0\|_2 \geq \frac{1}{h_1(mn \log n)^{1/d}}\right) \geq \left(1 - \frac{v_d}{mnh_1^d \log n}\right)^n \geq 1 - \frac{v_d}{mh_1^d \log n}.$$

Hence, the target quantity $\sum_{i=1}^m \mathbb{1}(E_i^c)$ is a Binomial random variable with number of observations $m$ and $\mathbb{P}(E_i^c) \leq v_d/(mh_1^d \log n)$. Using the first inequality of Lemma 4 with $\delta = 1, \lambda = m\mathbb{P}(E_i^c)$ gives

$$\mathbb{P}\left(\sum_{i=1}^m \mathbb{1}(E_i^c) \geq \frac{2v_d}{h_1^d \log n}\right) \leq \exp\left(-\frac{v_d}{3h_1^d \log n}\right).$$

Therefore, by using $h_1 \leq c(\log n)^{-2/d}$ with a small enough $c > 0$, the claimed results hold for sufficiently large $n$. $\qquad\square$

By Lemma 6 we know that $|I| \leq 1/(3h_1^d)$ with probability at least $1 - n^{-3}$. We also define the set of *good* indices as follows:

$$I^\star = \left\{i \in [m] : \min_{j \in I} \|X_i^1 - X_j^1\|_2 > \frac{h_1}{2}\right\}.$$

We claim that for any $i \in I^\star$, we have $|\tau(X_i^1)| = \widetilde{\Omega}(h_1^{\beta_\tau})$. In fact, for any $j \notin I$, the definition of $I$ gives

$$\|X_i^1 - X_j^1\|_2^{\beta_\tau} + \min_{k \in [n]} \|X_j^1 - X_k^0\|_2^{\beta_\mu} \geq \min_{k \in [n]} \|X_j^1 - X_k^0\|_2^{\beta_\mu} > \left(\frac{1}{h_1(mn \log n)^{1/d}}\right)^{\beta_\mu} = \widetilde{\Omega}(h_1^{\beta_\tau}),$$

where the last identity follows from the choice of $h_1$. As for the other case $j \in I$, the definition of $I^\star$ gives

$$\|X_i^1 - X_j^1\|_2^{\beta_\tau} + \min_{k \in [n]} \|X_j^1 - X_k^0\|_2^{\beta_\mu} \geq \|X_i^1 - X_j^1\|_2^{\beta_\tau} \geq \min_{j \in I} \|X_i^1 - X_j^1\|_2^{\beta_\tau} = \Omega(h_1^{\beta_\tau}),$$

as claimed. Hence, all $|\tau(X_i^1)|$ with index $i \in I^\star$ have a large magnitude, and it remains to show that $|I^\star|$ is large with high probability. In fact, conditioning on $|I| \leq 1/(3h_1^d)$, the following set

$$A := \left\{x \in [0,1]^d : \min_{j \in I} \|X_i^1 - x\|_2 \leq \frac{h_1}{2}\right\}$$

has $d$-dimensional volume at most $|I| \cdot h_1^d \leq 1/3$. Note that $X_i^1$ is uniformly distributed, the region $A$ has probability at most $1/3$. Hence, by the standard Binomial tail bound (cf. Lemma 4), we conclude that with probability at least $1 - 2n^{-3}$, the number of treatment covariates $X_i^1$ falling into $A$ is at most $m/2$. Consequently, we have $|I^\star| \geq m/2$. In other words, with high probability, at least a constant fraction of $X_i^1$ satisfies $|\tau(X_i^1)| = \widetilde{\Omega}(h_1^{\beta_\tau})$, and therefore $\|\tau\|_1 = \widetilde{\Omega}(h_1^{\beta_\tau})$ with high probability as well. Now the two-point method applied to $(\mu_0, \tau)$ and $(\mu_0' \equiv 0, \tau' \equiv 0)$ together with the choice of $h_1$ gives the desired lower bound $\widetilde{\Omega}((\kappa/n^2)^{1/d(\beta_\mu^{-1} + \beta_\tau^{-1})})$.

Next we prove the final lower bound $\widetilde{\Omega}((\kappa\sigma^2/n^2)^{1/(2+d(\beta_\mu^{-1} + \beta_\tau^{-1}))})$. To this end, we construct exponentially many hypotheses $(\mu_0^v, \tau^v)$ indexed by $v \in \{\pm 1\}^M$ and apply Fano's inequality. Specifically, let $h_1, h_2$ be the bandwidths given as follows:

$$h_1 = \widetilde{\Theta}\left(\frac{1}{(mn)^{1/d}}\left(\frac{\sigma}{\sqrt{mn}}\right)^{-\frac{2\beta_\tau}{2\beta_\mu\beta_\tau + d(\beta_\mu + \beta_\tau)}}\right),$$

$$h_2 = \widetilde{\Theta}\left(\left(\frac{\sigma}{\sqrt{mn}}\right)^{\frac{2\beta_\mu}{2\beta_\mu\beta_\tau + d(\beta_\mu + \beta_\tau)}}\right).$$

and $M = h_2^{-d}$. Fix any smooth function $g$ supported on $[0, 1]^d$, then the standard dilation analysis yields that

$$\tau^v(x) = c \cdot h_2^{\beta_\tau} \sum_{i=1}^{M} v_i g\left(\frac{x - x_i}{h_2}\right)$$

for sufficiently small constant $c > 0$ belongs to the Hölder ball $\mathcal{H}_d(\beta_\tau)$ for all $v \in \{\pm 1\}^M$, where $x_1, \cdots, x_M$ are vertices of the small cubes so that $\{x_1, \cdots, x_M\} + [0, h_2]^d = [0, 1]^d$. Note that

$$\|\tau^v - \tau^{v'}\|_1 = ch_2^{\beta_\tau}\|g\|_1 \cdot d_{\mathrm{H}}(v, v'),$$

where $d_{\mathrm{H}}(v, v') = M^{-1} \sum_{i=1}^{M} \mathbb{1}(v_i \neq v'_i)$ is the normalized Hamming distance between binary vectors $v$ and $v'$. By the Gilbert-Varshamov bound, there exists $\mathcal{V}_0 \subseteq \{\pm 1\}^M$ with $|\mathcal{V}_0| \geq 2^{M/8}$ such that $d_{\mathrm{H}}(v, v') \geq 1/5$ for all $v, v' \in \mathcal{V}_0$. Let $V$ be a random variable uniformly distributed in $\mathcal{V}_0$, and $Y$ be the collection of all outcomes in (1) based on $\tau^v$ and $\mu_0^v$ defined below. The standard Fano-type arguments (see, e.g. [Tsy09]) give

$$R_{n,d,\beta_\mu,\beta_\tau,\sigma}^{\mathrm{random}}(\kappa) = \Omega\left(h_2^{\beta_\tau}\left(1 - \frac{I(V; Y) + \log 2}{M/8}\right)\right).$$

Next we give an explicit construction of $\mu_0^v$ for each $v \in \{\pm 1\}^M$, and upper bound the mutual information $I(V; Y)$.

The baseline functions $\mu_0^v$, also indexed by $v \in \{\pm 1\}^M$, are chosen to offset the differences of $\tau^v$ on the treatment covariates. Motivated by Lemma 2, we choose the reference function

$$\tilde{\mu}_0(X_i^1) = c \cdot \min_{j \in [m]}\left(\|X_i^1 - X_j^1\|_2^{\beta_\mu} + \min_{k \in [n]}\|X_i^1 - X_k^0\|_2^{\beta_\mu}\right),$$

and

$$\mu_0^v(X_i^1) = -\mathrm{sign}(\tau^v(X_i^1)) \cdot \left(|\tau^v(X_i^1)| \wedge \tilde{\mu}_0(X_i^1)\right),$$

and $\mu_0^v(X_i^0) = 0$ for any $i \in [n]$. As before, using similar arguments we conclude that the above value specifications satisfy the conditions of Lemma 2, and thus each $\mu_0^v$ can be extended to a function in $\mathcal{H}_d(\beta_\mu)$ for each $v$. Since distributions of the control outcomes do not depend on $v$, $\mu_0^v(X_i^1) + \tau^v(X_i^1) = 0$ whenever $\tilde{\mu}_0(X_i^1) \geq ch_2^{\beta_\tau}\|g\|_\infty \geq |\tau^v(X_i^1)|$, and all noises are normal distributed with variance $\sigma^2$, we have

$$I(V; Y) \leq \max_{v \neq v' \in \mathcal{V}_0} D_{\mathrm{KL}}(P_{Y|V=v}\|P_{Y|V=v'})$$

$$= \max_{v \neq v' \in \mathcal{V}_0} \sum_{i=1}^{m} \frac{(\mu_0^v(X_i^1) + \tau_0^v(X_i^1) - \mu_0^{v'}(X_i^1) - \tau_0^{v'}(X_i^1))^2}{2\sigma^2}$$

$$\leq 4c^2 h_2^{2\beta_\tau}\|g\|_\infty^2 \cdot \sum_{i=1}^{m} \mathbb{1}(\tilde{\mu}_0(X_i^1) \leq ch_2^{\beta_\tau}\|g\|_\infty).$$

By the choices of $h_1$ and $h_2$ above, we have $h_2^{\beta_\tau} = \widetilde{\Omega}((h_1(mn)^{1/d})^{-\beta_\mu})$. Hence, similar arguments of Lemma 6 lead to

$$\left|\left\{i \in [m] : \min_{j \in [n]}\|X_i^1 - X_j^0\|_2 \wedge \min_{j \neq i}\|X_i^1 - X_j^1\|_2 \leq \frac{1}{h_1}\left(\frac{1}{mn\log n}\right)^{\frac{1}{d}}\right\}\right| = O(h_1^{-d})$$

with probability at least $1 - n^{-3}$. Also, for any $i \in [m]$ not belonging to the above set of indices, it is straightforward to see that $\tilde{\mu}_0(X_i^1) = \widetilde{\Omega}((h_1 n^{2/d})^{-\beta_\mu})$. Hence, with probability at least $1 - n^{-3}$ we have $\sum_{i=1}^{n} \mathbb{1}(\tilde{\mu}_0(X_i^1) \leq ch_2^{\beta_\tau}\|g\|_\infty) = O(h_1^{-d})$. Therefore,

$$I(V; Y) = O\left(\frac{h_2^{2\beta_\tau}}{h_1^d \sigma^2}\right) = O\left(h_2^{-d}\right) = O(M)$$

holds with high probability over the random covariates. Plugging the above upper bound of the mutual information into the Fano's inequality, we conclude that

$$R_{n,d,\beta_\mu,\beta_\tau,\sigma}^{\mathrm{random}}(\kappa) = \widetilde{\Omega}(h_2^{\beta_\tau}) = \widetilde{\Omega}\left(\left(\frac{\kappa\sigma^2}{n^2}\right)^{\frac{1}{2+d(\beta_\mu^{-1}+\beta_\tau^{-1})}}\right),$$

as desired.

## C Proof of Main Lemmas

### C.1 Proof of Lemma 1

We first establish the following inequality: for any $\varepsilon > 0$, there exists a constant $C_d > 0$ depending only on $d$ such that

$$\int_{[0,1]^d} f(x)\mathbb{1}(m[f](x) \leq \varepsilon)dx \leq C_d\varepsilon. \tag{15}$$

To establish (15), we apply the maximal inequality in Lemma 5 to the set $A = [-\sqrt{d}, 1 + \sqrt{d}]^d$ and measures $\mu_1(dx) = f(x)\lambda(dx)$, $\mu_2(dx) = \lambda(dx)$, with $\lambda$ being the $d$-dimensional Lebesgue measure. Then clearly for every $x \in [0,1]^d$, we have

$$\sup_{\rho>0:B(x;\rho)\subseteq A} \left( \frac{\mu_2(B(x;\rho))}{\mu_1(B(x;\rho))} \right) \geq \frac{1}{m[f](x)}.$$

Consequently, choosing $t = 1/\varepsilon$ in Lemma 5 gives

$$\int_{[0,1]^d} f(x)\mathbb{1}(m[f](x) \leq \varepsilon)dx \leq \mu_1\left( \left\{ x \in A : \sup_{\rho>0:B(x;\rho)\subseteq A} \left( \frac{\mu_2(B(x;\rho))}{\mu_1(B(x;\rho))} \right) \geq \frac{1}{\varepsilon} \right\} \right)$$
$$\leq C_d'\varepsilon \cdot \mu_2(A) = C_d\varepsilon,$$

establishing (15).

Next we make use of (15) to prove Lemma 1. For $\lambda > eC_d$, let $\varepsilon$ be a random variable following an exponential distribution with rate $\lambda$. Then clearly $\mathbb{P}(m[f](x) \leq \varepsilon) = \exp(-\lambda \cdot m[f](x))$, and taking expectation at both sides of (15) leads to

$$\int_{[0,1]^d} f(x)\exp(-\lambda \cdot m[f](x))dx \leq C_d \cdot \mathbb{E}[\varepsilon] = \frac{C_d}{\lambda},$$

as claimed. For $\lambda \leq eC_d$, note that applying the above arguments gives that for any $t \geq 1$,

$$\int_{[0,1]^d} f(x)\exp(-t\lambda \cdot m[f](x))dx \leq \frac{C_d}{t\lambda}.$$

Hence, using the fact that $f$ is a density and the Hölder's inequality, we have

$$\int_{[0,1]^d} f(x)\exp(-\lambda \cdot m[f](x))dx \leq \inf_{t \geq 1} \left( \int_{[0,1]^d} f(x)\exp(-t\lambda \cdot m[f](x))dx \right)^{\frac{1}{t}}$$
$$\leq \inf_{t \geq 1} \left( \frac{C_d}{t\lambda} \right)^{\frac{1}{t}} = \exp\left( -\frac{\lambda}{eC_d} \right),$$

where the minimizer in the last step is $t = eC_d/\lambda \geq 1$.

### C.2 Proof of Lemma 2

The necessity result simply follows from the Hölder ball condition, and it remains to prove the sufficiency. Without loss of generality we assume that $L = 1$. We first show that given $\{(x_i, y_i)\}_{i \in [n]}$ satisfying the condition of Lemma 2, for any other $x \in \mathbb{R}^d \setminus \{x_i\}_{i \in [n]}$ there exists a value specification $y \in \mathbb{R}$ of $f(x)$ such that the same condition holds for the $(n+1)$ points $\{(x_i, y_i)\}_{i \in [n]} \cup \{(x, y)\}$, i.e.

$$|y_i - y| \leq \|x_i - x\|_2^\beta, \quad \forall i \in [n].$$

To prove the existence of $y$, it suffices to show that

$$\min_{i \in [n]} \left\{ y_i + \|x_i - x\|_2^\beta \right\} \geq \max_{j \in [n]} \left\{ y_j - \|x_j - x\|_2^\beta \right\}.$$

In fact, for $i, j \in [n]$,

$$\left(y_i + \|x_i - x\|_2^\beta\right) - \left(y_j - \|x_j - x\|_2^\beta\right) = (y_i - y_j) + \|x_i - x\|_2^\beta + \|x_j - x\|_2^\beta$$

$$\geq -\|x_i - x_j\|_2^\beta + \|x_i - x\|_2^\beta + \|x_j - x\|_2^\beta$$

$$\geq 0.$$

Hence, we may construct the $(n+1)$-th point based on $n$ points, and repeating this procedure leads to a construction of $f$ on any countable dense subset $\mathcal{B} \subseteq \mathbb{R}^d$. Now the proof is completed by taking

$$f(x) = \lim_{x_n \in \mathcal{B}, x_n \to x} f(x_n).$$

### C.3 Proof of Lemma 3

For $t = 0$, the lemma is obvious. In the following we consider $t \geq 1$ and write the linear equations in the matrix form

$$\begin{bmatrix} 1 & 1 & \cdots & 1 \\ z_0 & z_1 & \cdots & z_t \\ \vdots & \vdots & \vdots & \vdots \\ z_0^t & z_1^t & \cdots & z_t^t \end{bmatrix} \begin{bmatrix} w_0 \\ w_1 \\ \vdots \\ w_t \end{bmatrix} = \begin{bmatrix} 1 \\ 0 \\ \vdots \\ 0 \end{bmatrix}.$$

where $z_j = x - x_j, 0 \leq j \leq t$. Since the above square matrix is a Vandermonde matrix, the weight vector $(w_0, \cdots, w_t)$ exists and is unique. Now by the Cramer's rule, for $i \geq 1$ we have

$$|w_i| = \frac{\det \begin{bmatrix} 1 & \cdots & 1 & \cdots & 1 \\ z_0 & \cdots & 0 & \cdots & z_t \\ \vdots & \vdots & \vdots & \vdots & \vdots \\ z_0^t & \cdots & 0 & \cdots & z_t^t \end{bmatrix}}{\det \begin{bmatrix} 1 & \cdots & 1 & \cdots & 1 \\ z_0 & \cdots & z_i & \cdots & z_t \\ \vdots & \vdots & \vdots & \vdots & \vdots \\ z_0^t & \cdots & z_i^t & \cdots & z_t^t \end{bmatrix}} = \prod_{0 \leq j \leq t, j \neq i} \left| \frac{z_j}{z_j - z_i} \right|$$

$$= \frac{\Delta}{|z_0 - z_i|} \cdot \prod_{1 \leq j \leq t, j \neq i} \left| \frac{z_j}{z_j - z_i} \right| \leq m\Delta \cdot (t+1)^{t-1},$$

where the last inequality follows from $|z_j| \leq (t+1)/m$ and $|z_j - z_i| \geq 1/m$ for $i \neq j$. The proof is completed by noting that

$$|w_0| \leq 1 + \sum_{i=1}^{t} |w_i| \leq 1 + m\Delta \cdot t(t+1)^{t-1} \leq 1 + t(t+1)^{t-1} =: C.$$