[Reviews · NeurIPS 2020]

Review 1

Summary and Contributions: The paper theoretically studies nonparametric minimax-optimal rates for heterogeneous treatment effect estimation. The main assumption is that the conditional average treatment effect function is smoother than the treatment-specific baseline responses. ` ---- I have read the author feedback. Thanks for the clarifications. The additional experiments are particularly illuminating.

Strengths: soundness of the claims: theoretically-grounded lower bounds for nonparametric estimation relevance: heterogeneous treatment effects are well-studied

Weaknesses: Relevance: The minimax bound in this tau-smoother-than-baseline setting is relevant is (but perhaps less so because it does not take into account additional structure on the propensity scores, which typically algorithmic approaches for causal inference do) The proposed algorithm seems relevant mostly because it is a constructive procedure to achieve the bound, rather than providing additional insight on how this theoretical understanding should affect algorithm design for heterogeneous treatment effects.

Correctness: The claims are correct to the best of my knowledge.

Clarity: The paper is reasonably well written. (It is useful to illustrate the simplification for fixed design)

Relation to Prior Work: Comparison to the rates achieved by previous work and different assumptions made could greatly improve relevance. For example, the following paper studied minimax bounds for heterogeneous treatment effect estimation: Alaa, Ahmed, and Mihaela Schaar. "Limits of estimating heterogeneous treatment effects: Guidelines for practical algorithm design." International Conference on Machine Learning. 2018. Though they may not have made different structural assumptions on \tau rather than \mu, they focus a bit on algorithmic implications of the work.

Reproducibility: Yes

Additional Feedback: Some questions: Re: line 295-296 Could you comment on why this is the case -- is this purely a consequence of the random design focus? Since intuitively the order of generating P(X) P(T|X) vs. P(T) P(X|T) shouldn't matter (these are equivalent factorizations of the joint distribution) Assumption 1: When is it expected for the inequality to hold strictly? Clearly one example where this is the case is when \tau is constant. Are there other examples where it holds strictly? Intermediate/realistic settings seem they might be better modeled by assuming that "\tau" is well-approximated by a function with a strictly smaller smoothness parameter".


Review 2

Summary and Contributions: - This paper studies the estimation of heterogeneous treatment effects (HTE) using nonparametric methods. In particular, the authors give results on the minimax rate with which HTE can be learned, under assumptions on smoothness of the true effect functions, and propose algorithms for doing so. Two settings are considered: a fixed design where subjects in different treatment groups are essentially matched on covariates. In the second setting, a random design, covariates are sampled from a distribution. Finally, the proposed algorithms for each setting are shown to achieve the minimal optimal rates. After author response: I maintain my position that this is a good paper that would be a nice contribution to the conference.

Strengths: - The paper studies a problem---causal effect estimation---that is highly relevant for many applications of machine learning. It provides a valuable addition to the theoretical understanding of this problem. In particular, while many works are agnostic to the distribution of treatments and covariates, this paper studies two distinct and concrete settings which are relevant in practice. - This choice to study concrete settings makes interpretation of the results more intuitive and the take-aways more clear. This is illustrated in the discussion of the nearest-neighbor-based estimator which discards observations that are poorly matched. To add to this, the authors provide a useful discussion about the limitations of the proposed approaches at the end of the paper. - A comprehensive survey of related work is given and comparisons with theoretical results from these are made.

Weaknesses: - The two settings considered, the fixed design and the low-smoothness setting are both fairly restricted. In particular, requiring that the smoothness parameter beta < 1 is rather strong, as indicated by the example/discussion given in Section 4. - The machinery used for analysis, e.g., kernel-methods and differencing are known and used often in nonparametric estimation. Nevertheless, the application yields interesting results here. - There are no empirical results included in the paper. These could have been used to study the conjectured phase transition from beta < 1 to beta > 1. Given the proposed algorithms, this seems like a missed opportunity.

Correctness: - The results of the paper appear correct.

Clarity: - The goals of the paper are clearly stated and followed up on, see e.g. ln 78: "The main aim of this paper is to characterize the tight minimax rates for the above quantities". The paper is well written and assumptions are clearly stated.

Relation to Prior Work: - Good overview of related work

Reproducibility: Yes

Additional Feedback:


Review 3

Summary and Contributions: The author calculated a minimax rate of CATE functions under the setting, CATE is smoother than the baseline function. They consider two designs and find the minimax rate and the method nearly achieving it.

Strengths: This paper is quite strong a paper. First, this paper searches for the fundamental limitation (minimax rates) of CATE estimation. I do not think the rate result is obvious. So, it is technically challenging. At the same time, minimax rates have a reasonable interpretation well explained in the paper. Second, the method (nearly) achieving it is also novel in at least a causal inference community.

Weaknesses: I do not think there is a big weakness. One thing the reader would want to know is the practical performance (experiment) of the proposed methods. I recommend the author to add it so that people can see the implication of the theorems.

Correctness: I am familiar with causal inference literature. But I know the basics of minimax theory. In this sense, I am not sure these theorems are really correct. Based on my educated guess and their intuitive explanation (and my brief checking of the proof), it looks correct.

Clarity: Yes. It is clearly written. The author tries to convey a difficult theorem to the reader in an easier way.

Relation to Prior Work: yes. It is clearly discussed.

Reproducibility: Yes

Additional Feedback:


Review 4

Summary and Contributions: The purpose of this paper is to provide new theoretical tools and bounds for the heterogeneous treatment effect (HTE) estimation in causal inference. This work is in line with a fairly current theme: the HTE estimation is experiencing a growing interest in applications, particularly in the field of personalized medicine. To avoid strong assumptions and to benefit from a broader scope of application, the authors focus on nonparametric estimation. As the authors point out, much effort has been devoted to proposing practical methods, but not so much to the statistical study of nonparametric HTE estimation. This paper establishes minimax rates with dependence on both the geometry of the covariates, and parameters related to propensity scores and noise levels. The authors provide two designs: a fixed and a random one. In the fixed design, the covariates are generated from the same regular grid, translated by a vector able to quantify the matching distance between the control and treatment groups. The random design is more realistic, with no matching parameters and using the propensity score. Section 2 is devoted to the study of the fixed design. In this design, the estimation relies on the regular structure of the grid and kernel estimator. Characterize the minimax L2 risks for the random design (Section 3) is a more tricky problem. Here, the authors propose a two-stage nearest-neighbor estimator and show that the minimax estimation error exhibits three different behaviors (Theorem 2).

Strengths: The article is well-written, the explanations are clear and detailed. In particular, the different contributions regarding the existing literature are easily identifiable. All the theorems are put into context, and the different bounds obtained are explained, effectively highlighting the key ideas.

Weaknesses: Detailed algorithms are provided, but I would have liked to see them "in action". However, I am aware that the size constraint does not necessarily allow me to propose numerical studies and I find the size of the article well mastered as it is. A really minor concern: I agree that Theorems 1 and 2 are direct consequences of Theorems 3 and 4 and 5 and 6 respectively, as clearly stated in the appendices. I just find it a little unfortunate that we have to refer to the appendices to read it, I would have appreciated a mention in the main document.

Correctness: As far as I checked, the proofs are correct, detailed and easy to follow. The decomposition into lemmata is judicious.

Clarity: Good readability and organization. The annexes are also well written. The notation could be slightly improved as it is sometime difficult do notice the dot above the “=” sign (introduced line 138).

Relation to Prior Work: I find the paper well put in context.

Reproducibility: Yes

Additional Feedback: Post-rebuttal: I thank the authors for their detailed response, which confirms my positive opinion on this work.

[Author Response · NeurIPS 2020]

We are grateful to the reviewers for the valuable comments. Below we respond to each and every point of the reviewers.

**Reviewer 1**: We thank the reviewer for the comments. Please find our point-to-point response below.

*Propensity score*: Our work does consider a mild constraint on the propensity score (encoded by the likelihood ratio
$\kappa$ in Assumption 2), and our minimax risks also have tight dependence on $\kappa$. We agree that other modelings of the
propensity score are also possible (e.g. modeled as another nonparametric function), but for size constraint we decide to
leave it as future work.

*Guideline on algorithm design*: While we agree that the main focus of our work is to characterize the tight minimax rates
in theory, we also provide several practical insights. First, estimating two baselines separately and differencing is no
longer optimal when the HTE is smoother, and one should estimate the HTE directly based on the pseudo-observations
obtained via covariate matching. Second, the covariate geometry plays a central role in HTE estimation, and one should
discard observations with poor matching quality. Their usefulness is illustrated in the empirical results below.

*Comparison with Alaa and Schaar'18*: As correctly claimed by the reviewer, the main difference is on the assumption:
they imposed smoothness constraints on two baselines, while we assume a smoother HTE. This difference leads to
several consequences. First, one of their main take-aways is that the minimax rate is determined by the less smooth
baseline, while in our case it is determined by an interpolation of both smoothness parameters depending on the
covariate geometry. Second, we also provide new practical guidelines to employ the covariate geometry and perform
covariate matching (see above), a point which is unnecessary in [AS18] where the baselines are estimated separately.
We cited this work in Line 121-123 of the submission, and will elaborate more on the differences in the final paper.

*Line 295-296*: Thanks for pointing out this statement which is not very precise. In our work, we study the minimax risk
after conditioning on both random realizations of the covariates and group assignments, meaning that the worst-case
HTE can depend on both the above realizations. In contrast, [NW17] only conditions on the random covariates and
considers an expected risk taken with respect to the randomness in the group assignments, where the worst-case HTE
cannot depend on the group assignments. We will further clarify this point in the final paper.

*Assumption 1*: This assumption is a nonparametric modeling of the fact/belief that the HTE is simpler than the baselines.
For example, in (semi)parametric modelings the HTE is typically assumed to be a constant or a low-dimensional/sparse
linear function; our assumption is its nonparametric counterpart. We agree that we could also assume the HTE to be
*approximated* by a smoother function, but without additional assumptions, this approximation error will enter the play
as an additive error after a simple use of the triangle inequality. So for brevity we assume the HTE itself is smoother.

**Reviewer 2**: We thank the reviewer for the comments. Please find our point-to-point response below.

*Fixed grid and smoothness constraint*: We agree that the fixed design assumption is not very practical, and we include it
mainly to illustrate how the combination of the smoothness and the covariate geometry determines the final minimax
rate (cf. Line 62-63 of the submission). We also agree that restricting to smoothness less than $1$ is unfortunate for us in
theory, but we could provide three reasons for this. First, capturing higher order smoothness could be a challenging task
in nonparametric problems, and there are a number of other examples where how to solve this task is open. Second,
there are insights suggesting that the cases $\beta > 1$ and $\beta < 1$ exhibit fundamentally different behaviors in the minimax
rate. Third, there seems to be a fundamental approximation-theoretic difficulty when arguing the lower bounds for
$\beta > 1$. We refer to Line 267-287 of the submission for details.

*Well-known machinery*: We agree that the kernel or nearest-neighbor tools are well-known in nonparametric methods.
Nevertheless, our main aim is to provide a careful combination of these tools which also takes the covariate geometry
into account, and show that the aforementioned combination (nearly) attains the minimax rates for HTE estimation.

*Empirical results*: Thank you (and also Reviewers 3-4) for bringing up this useful suggestion. We will add an extensive
collection of numerical experiments to the final paper, and we plot one example here to illustrate the performance
of our algorithm. Here (a) plots an oscilating baseline function $\mu_0$ with $d = 1$, and (b) plots a smoother HTE $\tau$.
We compare 4 estimators: *selected matching* as used in the paper, *full matching* without discarding any observation,
*kNN differencing* and *kernel differencing* which apply different methods to estimate both baselines and then take the
difference. Figure (b) plots one random instance of all estimators with sample size $n = 1,000$, and (c) plots the root
mean squared error (RMSE) as a function of sample sizes. We observe that utilizing covariate matching performs better
than estimate-then-difference, and throwing away covariates with poor matching quality also leads to a smaller error.

(a) Baseline function $\mu_0$.　　　(b) HTE $\tau$.　　　(c) RMSE vs sample size.

**Reviewer 3**: We thank the reviewer for the comments. We will add numerical experiments and results to the final paper
to illustrate the empirical performance and practical implications; please refer to our response to Reviewer 2 above.

**Reviewer 4**: We thank the reviewer for the comments. We will add numerical experiments and results to the final paper
to illustrate the algorithms in action; please refer to our response to Reviewer 2 above.

*Response to minor comments*: The complete statements of Theorems 3-6 are in the main document, and only the proofs
are relegated to the appendix. Meanwhile, we also provide the high-level proof ideas for these statements in the main
document. We will also replace the dot notation by tilde notation to gain more visibility - thanks for pointing out this.

[Meta-Review · NeurIPS 2020]

The knowledgeable reviewers agree that this is a good paper that warrants acceptance. There was no major concern raised during the discussion phase and the rebuttal has further supported the vote for acceptance. The paper is therefore accepted as a spotlight. If the authors have time and agree that it will be beneficial, the important improvement for the paper is to add a simple experiment that demonstrates the effectiveness of the proposed estimator.